# SELF-SUPERVISED LEARNING WITH SIDE INFORMATION

## ABSTRACT

A core assumption behind many successful self-supervised learning (SSL) methods is that different views of the same input share the information needed for downstream tasks. However, this MultiView assumption can be overly permissive in real-world settings, where task-irrelevant features may persist across views and become entangled with useful signals. Motivated by challenges in colonoscopy—where polyp cues must be isolated from dominant but irrelevant background textures—we present an information-theoretic analysis of this general failure mode in SSL. We further formalize this with our proposed Nuisance-Free MultiView (NF-MV) assumption, which reframes the goal of SSL as learning representations that are sufficient for task-relevant information while being invariant to shared nuisance structure. We theoretically show that such representations yield improved generalization, and derive an idealized objective that balances standard view alignment with a mutual information penalty on nuisance content. To implement this in practice, we introduce a method that leverages side information—auxiliary data that shares nuisance structure but does not contain any task-relevant signals. The nuisance penalty is then approximated using a Jensen-Shannon divergence between main and side representations, in a way that is tractable and compatible with standard joint embedding architectures. Experiments on synthetic tasks with spurious correlations and on real-world colonoscopy datasets demonstrate that the proposed method improves generalization for a wide range of SSL methods and architectures by learning the relevant features. These findings highlight the benefits of explicitly modelling what should not be preserved during self-supervised learning, offering a new and practical perspective on the MultiView framework.

## 1 INTRODUCTION

Machine learning and deep learning are rapidly transforming medical image analysis, offering promising avenues to improve diagnostic accuracy and efficiency across numerous clinical applications. Among the applications that can benefit significantly from these advances is the detection of colorectal cancer (CRC), a major global health concern with approximately two million new cases detected annually (Morgan et al., 2023). Most CRCs originate from adenomatous polyps, whereas hyperplastic polyps pose limited risk of transitioning to cancer (Bretthauer et al., 2022). Despite its importance, colonoscopy remains highly operator-dependent, and variations in visual perception and clinical skill can reduce the effectiveness of screenings (Cherubini & East, 2023). AI-based systems have been proposed to assist in polyp detection and classification, but they typically rely on large-scale labelled datasets — which are costly and time-consuming to obtain. Self-supervised learning (SSL) offers a promising alternative by enabling models to learn useful representations from unlabeled data. Some of the most successful SSL approaches are joint embedding architectures (JEAs), which align representations of augmented views of the same input. These methods are motivated by the *MultiView assumption* (Sridharan & Kakade, 2008): the relevant information is shared across augmented views, and aligning these views encourages the encoder to learn useful representations. Modern JEAs, such as SimCLR, Barlow Twins, and Masked Siamese Networks,

have achieved outstanding results relying on this assumption (Chen et al., 2020; Bardes et al., 2022; Assran et al., 2022; 2023; Hu et al., 2024; Wang et al., 2023; Hirsch et al., 2023)[1].

However, the MultiView assumption can be overly permissive. It does not distinguish between task-relevant and task-irrelevant (nuisance) information that may be shared across views. In settings such as colonoscopy, augmented views often preserve for instance strong background textures, irrelevant to downstream diagnostic tasks. Standard SSL methods may entangle such nuisance features with the more subtle task-relevant signals, degrading downstream performance. To address this, we introduce the *Nuisance-Free MultiView* (NF-MV) assumption, an information-theoretic perspective on the MultiView setting that explicitly excludes shared nuisance structure from the representation. Under NF-MV, we frame the goal of SSL as learning representations sufficient for the task while being invariant to nuisance information (see Fig. 1). We implement this framework using *side information*—auxiliary data that shares nuisance structure but lacks task-relevant information—and penalize representational overlap using a Jensen-Shannon divergence between main and side representations. This leads to a simple and general extension of standard joint embedding objectives. We evaluate our method on a controlled image classification task with synthetic spurious correlations and on real-world colonoscopy image analysis. Our approach leverages this typically overlooked redundancy to support more effective representation learning.

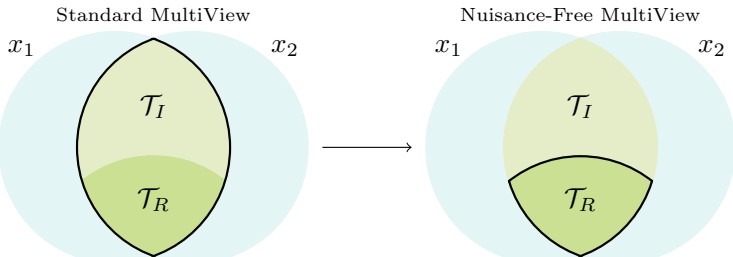

Figure 1: Illustration of information overlap between views $x_1, x_2$. Under the standard MultiView assumption (left), the learned representations encode features that support both task-relevant ($\mathcal{T}_R$) and task-irrelevant ($\mathcal{T}_I$) predictions. In contrast, our framework (right) leverages side information to promote representations that emphasize the task-relevant content.

## 2 BACKGROUND AND RELATED WORK

The Information Bottleneck (IB) framework (Tishby et al., 1999) provides a principled way to learn representations that are both compact and task-relevant. Given data $x$ and target $y$, the goal is to learn a stochastic mapping $p(z|x)$ that compresses $x$ into $z$ while preserving information about $y$. Chechik & Tishby (2002) extended this principle by introducing a nuisance variable $n$, modelling task-irrelevant structures. The goal is then to learn a representation $z$ that is informative about $y$ but invariant to $n$. Inspired by this extension, we propose leveraging side information in SSL by treating samples from an auxiliary side dataset $\mathcal{S}$ as exemplars of nuisance factors, and encourage the model to separate them from task-relevant signals learned from our main dataset.

**Side Information in Context.** Leveraging auxiliary datasets is an active area for research, previously explored in domain adaptation and contrastive analysis. Domain Adversarial Neural Networks employ a minimax problem where a domain discriminator tries to distinguish between source and target domains, while the feature extractor learns to produce domain-invariant features that minimize a classification loss on the source domain (Ganin et al., 2016; Long et al., 2018). Similarly, Domain Separation Networks (Bousmalis et al., 2016) decompose representations into shared and private components, preserving task-relevant information while isolating domain-specific variations. In

---

[1]Masking is prone to violate this assumption in certain domains. We hypothesise that this may explain why masking-based approaches sometimes underperform in medical domains, where adaptive masking strategies have proven useful (Yang et al., 2023; Basu et al., 2024; Hu et al., 2024).

contrast to Domain Adversarial/Separation Networks, our goal is to isolate and utilize the domain-specific signals as the useful representations. In addition, these methods assume that source and target domains share the same label spaces but differ in low-level statistics, an assumption that we do not make in our work. More closely related to our work is contrastive analysis (CA). CA methods assume access to a target dataset containing both salient and common (nuisance) variations and a background dataset that contains only common patterns. Their goal is to extract the target-specific variations by contrasting against the background features (Zou et al., 2013). This is achieved by using multiple generative encoders and mutual information penalties between the target and background encoders (Louiset et al., 2024a; Weinberger et al., 2022). These generative methods optimise log-likelihood objectives, and thus focus on modelling densities via the joint distribution: a strength for generation but typically less ideal for the discriminative structure needed in classification tasks, which our work focuses on. Most recently, SepCLR (Louiset et al., 2024b) employed deterministic encoders, combining CA with contrastive learning to learn the salient representations better suited for discriminative tasks. Importantly, however, CA-based techniques rely on multiple encoders and feature spaces, and thus incur substantial computational and memory costs, limiting scalability. This is particularly the case for momentum-based JEA architectures (Grill et al., 2020), which would require four separate encoders to allow the implementation of CA methods. Instead, our work targets separation within a single feature space using a single encoder, aiming at negligible computational overhead and easy integration with any existing JEA method.

## 2.1 Self-Supervised Learning

Self-Supervised Learning (SSL) employs self-designed signals to establish a proxy objective between the input and the signal, enabling training without additional labels. The model is initially trained using this proxy objective, and then fine-tuned on the target task. The training signals are derived from co-occurrence relationships within the data. To generate such signals, different kinds of architectures have been proposed, including generative models that reconstruct input data and Joint Embedding Architectures (JEAs). Joint Embedding Architectures process multiple views of an input signal through encoders to produce representations of the same underlying signal. The proxy objective is then to make these representations informative and mutually predictable, while avoiding trivial solutions by regularizing the feature space (Chen & He, 2021; Chen et al., 2020; He et al., 2020; Grill et al., 2020; Bardes et al., 2022). In this paper, we focus on JEA-based methods.

**Applications in Medical Imaging and Endoscopy.** SSL is set to become a key tool in medical and endoscopic image analysis. For instance, Wang et al. (2023) aligns spatiotemporal views to train encoders on endoscopy videos. Hirsch et al. (2023) applied the Masked Siamese Network approach to endoscopic video analysis, while $M^2CRL$ (Hu et al., 2024) combines contrastive learning and masked image modelling, achieving impressive results. These methods typically rely either on private datasets or curated clips that emphasise frames with visible polyps. For example, $M^2CRL$ leverages 10 publicly available datasets totalling over 33,000 videos and 5.5 million frames, but primarily focuses on sequences where non-polyp frames have been filtered out. In contrast, full-length colonoscopy videos are dominated by *negative* frames. The REAL-Colon dataset (Biffi et al., 2024), which we use for pre-training in our colonoscopy experiments, reflects this distribution: 87.6% of frames contain no polyps. Developing methods and frameworks that can effectively utilise this under-explored redundancy in real-world datasets has been a central motivation for our work.

## 2.2 SSL and the MultiView Assumption

The Information Bottleneck (IB) principle offers a foundational, information-theoretic framework to interpret supervised learning. However, adapting this principle to SSL remains challenging due to architectural and assumption-specific differences (Ziv & LeCun, 2024). Nevertheless, the MultiView assumption has been widely adopted to derive a family of IB-inspired methods (Wen et al., 2024; Huang et al., 2023; Federici et al., 2020; Tsai et al., 2021; Dubois et al., 2021).

**Assumption 1** (MultiView Assumption (Sridharan & Kakade, 2008)). *There exists an $\varepsilon > 0$ such that:*
$$I(y; x_2|x_1) \leq \varepsilon, \quad I(y; x_1|x_2) \leq \varepsilon$$
*In other words, different views $x_1, x_2$ of the same underlying sample $x$ do not contain substantially different information about the task label $y$; the views are assumed to share task-relevant content.*

The MultiView assumption implies that the information preserved across augmented views is task-relevant. Accordingly, alignment-based objectives used in SSL and JEAs are designed to promote invariance to the transformations used to generate the views - implicitly treating the shared content as sufficient for learning useful representations. Recent work has questioned the generality of this assumption and highlighted its limitations. Tian et al. (2020) showed that different types of augmentations are optimal for different tasks, suggesting that no single set of augmentations is universally effective. Wang et al. (2022) examined the case where not all task-relevant information is shared across views, and showed that representations learned via standard SSL may be insufficient under such conditions. These perspectives are complementary but opposite to ours. While these works examine the scenario in which the MultiView assumption is too strict, our focus is on the opposite case — when the assumption is too *permissive*. Specifically, we study cases where views share not only task-relevant signals but also task-irrelevant (nuisance) structures, which can degrade the quality of learned representations.

## 3 IRRELEVANT INFORMATION IN JOINT EMBEDDING ARCHITECTURES

In the MultiView SSL setting for JEAs we assume access to one unlabeled dataset $\mathcal{X}$, and some stochastic augmentation $A$. We define the set of paired views as $\mathcal{U} = \{(x_1^i, x_2^i)\}^K$ where $x_1^i, x_2^i \sim A(x^i)$ and $x^i \in \mathcal{X}$. By the MultiView assumption, the downstream tasks optimized during pre-training are those satisfying Assumption 1. We denote the set of these tasks by $\mathcal{T}$, which can be informally associated to the overlap between views in Fig. 1. More formally, the set of tasks are induced by:

$$\mathcal{T} = \{y \ : \ I(y; x_2|x_1) < \epsilon, \ I(y; x_1|x_2) < \epsilon\}, \quad \epsilon > 0. \tag{1}$$

Similar to Wang et al. (2022) we use the notion of sufficient representation and minimal sufficient representation. A representation $z_1^s$ of $x_1$ is sufficient for the other view $x_2$ if $I(z_1^s, x_2) = I(x_1, x_2)$, i.e. it keeps all shared information between $x_1, x_2$. Furthermore, a representation $z_1^{ms}$ of $x_1$ is minimal and sufficient if $I(z_1^{ms}, x_1) \leq I(z_1^s, x_1), \ \forall z_1^s$.

JEA architectures aim to optimize $I(z_1, z_2)$ to approximate $I(x_1, x_2)$. If the networks have enough capacity and sufficient data, the learned representations can be assumed sufficient. As the representations are learned by aligning the two views, they can also be considered minimal (Wang et al., 2022). By construction of $\mathcal{T}$, the representations are also minimal sufficient with respect to $\mathcal{T}$ [2]. However, it is often unnecessary—and potentially harmful—for representations to be useful for all tasks induced by the MultiView assumption. Let us divide the task set into relevant and irrelevant subsets, such that $\mathcal{T} = \mathcal{T}_R \cup \mathcal{T}_I$, where the relevant tasks $\mathcal{T}_R$ form a strict non-empty subset of $\mathcal{T}$. In this case, the representations learned by the JEA encoder are still sufficient for $\mathcal{T}_R$, but are no longer minimal with respect to it. We thus aim to learn representations that are minimal and sufficient for $\mathcal{T}_R$ alone, which leads to better generalization for tasks of actual interest. The advantage of doing so can be formalized by an adaptation of the Xu & Raginsky bound (Xu & Raginsky, 2017, Thm. 1).

**Theorem 1** (Generalization Benefit of Task-Specific Minimality). *Let $\mathcal{T}$ be a supervised learning task with distinct alphabet $\mathcal{Y}$ and let $\mathcal{T}' \subset \mathcal{T}$ be a strict sub-task. Let $Z = f(X)$ be minimal sufficient for $\mathcal{T}$ and $Z' = f'(X)$ be minimal sufficient for $\mathcal{T}'$. Draw a training set $S = (X_1, \ldots, X_n) \sim \mathcal{D}^n$, and let a fixed learning algorithm yield hypotheses $W = \mathcal{A}(Z^n)$ and $W' = \mathcal{A}((Z')^n)$. Suppose the loss $\ell(W, (X, \mathcal{T}'))$ is $\sigma$-sub-Gaussian. Then*

$$\mathbb{E}_{S,W}\big[\text{gen}(W, S)\big] \ \leq \ \sigma\sqrt{2\, I(Z; X)}, \qquad \mathbb{E}_{S,W'}\big[\text{gen}(W', S)\big] \ \leq \ \sigma\sqrt{2\, I(Z'; X)},$$

*so the upper bound for the generalisation error for $Z'$ is strictly tighter, as $I(Z'; X) < I(Z; X)$.*

This result suggests that it is preferable to use representations that are sufficient and minimal for the specific tasks of interest, rather than representations that are merely sufficient, as minimality lead to tighter generalization bounds. A formal proof and discussion can be found in the Appendix B.

**Nuisance Factors.** A key limitation of the MultiView assumption is that it does not distinguish between task-relevant and task-irrelevant (nuisance) information, as long as that information is shared across views. In realistic settings, shared but irrelevant factors often persist across augmentations and become entangled with the learned representation. These factors may be irrelevant or even

---

[2]Note that they are minimal w.r.t. the set of tasks, not for each individual task in $\mathcal{T}$.

harmful for the tasks of interest. Based on this, we propose a new perspective on the MultiView assumption. By defining what to consider as a nuisance, it is possible to control what the algorithm considers as relevant or irrelevant information. That is, the modeller specifies a structure $n$ that should be considered irrelevant. This nuisance specification induces a family of tasks for which the nuisance carries no label information.

**Assumption 2** (Nuisance-Free MultiView Assumption (NF-MV)). *Let $x_1, x_2$ be two views of an input $x$, and let $n_1, n_2$ be nuisance variables extracted from $x_1, x_2$, respectively. We assume:*

$$I(y; x_2 \mid x_1) \leq \varepsilon, \quad I(y; x_1 \mid x_2) \leq \varepsilon, \quad and \quad I(y; n_1) = I(y; n_2) = 0$$

*Then we say the Nuisance-Free MultiView assumption holds for $y$.*

If we substitute the MultiView assumption for the proposed Nuisance-Free MultiView Assumption, a new, strictly smaller, set of tasks arise.

**Definition 1** (NF-MV Induced Task Set). *Given nuisance $n$, we define the set of induced tasks as:*

$$\mathcal{T}_{nf}(n) := \{y \ : \ I(y; x_2 \mid x_1) \leq \varepsilon, \quad I(y; x_1 \mid x_2) \leq \varepsilon, \quad I(y; n) = 0\}$$

This task set consists of all labels that can be predicted equally well from either view *and* are independent of the nuisance. Once the modeller specifies a nuisance variable $n$, this isolates the subset of MultiView-induced tasks that are consistent with the modelling choice of what information should be ignored. If $n$ is sufficiently well-defined, then $\mathcal{T}_{nf}(n)$ captures the tasks for which the learned representations should be minimal and sufficient. This can be formalised as an idealized objective: $\max_\theta \ I(f_\theta(x_1), f_\theta(x_2)) - \gamma \ I(f_\theta(x), n)$, where $\gamma > 0$ is the parameter controlling the strength of nuisance suppression, and $x$ denotes a view of the input (either $x_1$ or $x_2$).

**Side Information to Define Nuisance.** The NF-MV assumption uses the existence of a nuisance variable $n$ that is independent of task-relevant information yet persists across views. In practice, such nuisance variables are not necessarily easy to express. To address this, we propose to approximate $n$ by using a side information dataset $\mathcal{S}$, containing samples that are structurally similar to the main data $\mathcal{X}$ but irrelevant to the tasks of interest. The assumption is that the nuisance structure is approximately captured by the overlap between $\mathcal{X}$ and $\mathcal{S}$. This perspective suggests an operational approach: define a binary indicator variable $B \in \{0, 1\}$ denoting the origin of a sample (main or side), and train the encoder to maximize the mutual information $I(z; B)$ instead of $I(f_\theta(x), n)$.

## 4 Leveraging Side Information via Jensen-Shannon Divergence

As motivated by the analysis above, it is preferred to learn an encoder that disentangles the nuisance features from relevant ones. To pinpoint nuisance structures we assume access to side information $\mathcal{S}$, that contains information that is (approximately) irrelevant but overlapping with the main dataset $\mathcal{X}$. The nuisance is then defined as the structural overlap between $\mathcal{X}$ and $\mathcal{S}$. When working with joint embedding models in a single feature space, there are additional subtleties to consider. First, we need to have informative representations of the side information $s \sim \mathcal{S}$ in order to disregard it. If the representations $f_\theta(s)$ are unreliable, it is not possible to disentangle the representations of the main data $f_\theta(x)$ between relevant and irrelevant structures. This means that we must use some of the representational power of the encoder to represent the irrelevant structures. Second, estimating and controlling mutual information in the extremely high-dimensional feature spaces where JEA methods operate is notoriously difficult. Estimators such as CLUB (Cheng et al., 2020) and L1Out (Poole et al., 2019) suffer from high variance and bias in these high-dimensional spaces. Moreover, since they require neural network parametrization, the training procedure becomes more complex.

**Estimating the Discrepancy via JSD.** Taking these considerations into account, we propose a simple objective for using side information with JEAs. Let $z = f_\theta(A(\omega))$, where $\omega \sim M_\alpha = \alpha \mathcal{X} + (1 - \alpha)\mathcal{S}$, and let $B_\alpha \in \{0, 1\}$ be the binary indicator with $\alpha = \mathbb{P}(B = 0)$. Maximizing the mutual information $I(z; B_\alpha)$ encourages the learned representations to retain information about whether it originated from $\mathcal{X}$ or $\mathcal{S}$, supporting the goal of disentangling nuisance from task-relevant structure. The mutual information $I(z; B_\alpha)$ can be expressed in closed form. A standard result from information theory shows that, when $\alpha = 0.5$, it holds that $I(z; B_{0.5}) = \text{JSD}(p(z \mid \mathcal{X}) \| p(z \mid \mathcal{S}))$.

This also holds more generally, for any $\alpha$, when considering a family of weighted Jensen-Shannon divergences (proof in Appendix A, Lemma 1). Specifically [3]:

$$I(z; B_\alpha) = \mathrm{JSD}_\alpha(p(z \mid \mathcal{X}) \| p(z \mid \mathcal{S})) = \alpha \, \mathrm{KL}(p(z \mid \mathcal{X}) \| M_\alpha) + (1 - \alpha) \, \mathrm{KL}(p(z \mid \mathcal{S}) \| M_\alpha), \quad (2)$$

where KL is the standard Kullback-Leibler divergence. This provides an estimator where the variance depends on the batch size instead of on the dimensionality of the feature space, and without any need for additional neural network parametrizations.

**Practical Computation.** In practice, the encoder and augmentations are potentially lossy and stochastic, so we consider the JSD as a tractable approximation to $I(z; B)$. We compute softmax predictions for each input and average them within each domain to estimate the empirical class distributions, effectively treating each output neuron as a prototype label:

$$\bar{z}_\mathcal{X} = \mathop{\mathbb{E}}_{x \sim \mathcal{X}}[\sigma(f_\theta(A(x)))], \quad \bar{z}_\mathcal{S} = \mathop{\mathbb{E}}_{s \sim \mathcal{S}}[\sigma(f_\theta(A(s)))], \quad \bar{z}_M = \mathop{\mathbb{E}}_{\omega \sim M_\alpha}[\sigma(f_\theta(A(\omega)))] \quad (3)$$

where $\sigma(\cdot) := \mathrm{Softmax}(\cdot)$, and evaluate the weighted divergence:

$$\mathrm{JSD}_\alpha\left(\bar{z}_\mathcal{X} \| \bar{z}_\mathcal{S}\right) = \alpha \, \mathrm{KL}(\bar{z}_\mathcal{X} \| \bar{z}_M) + (1 - \alpha) \, \mathrm{KL}(\bar{z}_\mathcal{S} \| \bar{z}_M), \quad (4)$$

A further motivation for this approximation arises by interpreting the softmax outputs as defining a discrete auxiliary variable $Y$. Given a representation $z$, we may view $\Pr(Y = y \mid \sigma(z))$ as a classifier-induced label distribution. Under this view, the batch-averaged softmax vectors $\bar{z}_\mathcal{X}$ and $\bar{z}_\mathcal{S}$ provide Monte Carlo estimates of the domain-conditional label distributions $\Pr(Y \mid B = 0)$ and $\Pr(Y \mid B = 1)$. It then follows that

$$I(Y; B) = \mathrm{JSD}_\alpha\left(\mathbb{E}[\sigma(z) \mid B = 0] \| \mathbb{E}[\sigma(z) \mid B = 1]\right).$$

By the data processing inequality, $I(Y; B) \leq I(Z; B)$, so the Jensen-Shannon divergence acts as a tractable lower bound on the mutual information we aim to maximize. This perspective provides an information-theoretic justification for our estimator: although coarse, it gives a reliable signal for separating relevant and nuisance structure in the learned representation. Importantly, this objective is straightforward to compute, introduces negligible overhead, and is architecturally agnostic, making it a simple and modular addition to a wide range of SSL methods.

## 5 EXPERIMENTS

We first conduct experiments in a controlled setting on natural images (using Cifar), showing that side information can mitigate bias learned during SSL pre-training. To demonstrate that our approach is not tied to any specific SSL method, we performed the experiments using Barlow Twins (Zbontar et al., 2021) and CorInfoMax (Ozsoy et al., 2022). Next, we perform experiments on real-world colonoscopy data. We pre-train both the baseline Masked Siamese Network (MSN) and our proposed side information-aware variant (MSN-SI) using a similar architecture to that employed by Hirsch et al. (2023). However, we use the public REAL-Colon dataset (Biffi et al., 2024), which comprises full-procedure colonoscopy videos, retaining the 87.6% frames that are polyp-negative.

**Baselines.** Throughout the experiments, we compare our method against two types of baselines. The *standard baselines* are models pre-trained on the main dataset, without access to any side information. For the *naive baselines* (-N), samples from the side dataset are added to the pre-training. The aim of this setting is to assess whether exposing the model to irrelevant structures is sufficient to encourage better representations. For the controlled experiments, we also compare with SepCLR Louiset et al. (2024b) from contrastive analysis.

### 5.1 CONTROLLED EXPERIMENTS

We construct two variants of a hybrid Cifar10+MNIST dataset: a correlated version (C-Cifar10) and an uncorrelated version (U-Cifar10). In both, MNIST (LeCun et al., 1998) digits are randomly scaled (0.5–1.0) and overlaid onto Cifar10 (Krizhevsky & Hinton, 2009) images. In C-Cifar10,

---

[3]It should be noted that this only holds exactly if the representation is lossless w.r.t. the source separation, which is not necessarily true.

Figure 2: The encoder is pre-trained on the biased data and the side information. Linear and k-NN classifiers are then trained on top of the frozen encoder using either the biased or the uncorrelated data. Evaluation is then performed on uncorrelated data in both cases.

each Cifar10 class is consistently paired with the MNIST digit of the same class (e.g., class 0 with digit 0), introducing a spurious correlation. In U-Cifar10, digits are assigned randomly. The classification target in the downstream task is the MNIST digit, making the background a task-irrelevant confounder. To introduce side information, we incorporate unlabelled samples from Cifar100 (Krizhevsky & Hinton, 2009), as it shares structure with the input but is unrelated to the MNIST classification task. During pre-training, a proportion $R_{\mathrm{SI}}$ of each batch consists of side samples. Our objective is to determine whether integrating side information during SSL pre-training enables the encoder to focus on task-relevant signals and disregard spurious correlations.

**Evaluation.** Representations are evaluated using two methods: linear probing (LP) and k-nearest neighbours (k-NN). For LP, a linear classifier is trained on top of the frozen encoder for 100 epochs using SGD with momentum 0.9 and no weight decay. We train the LP/k-NN on either U-Cifar10 (allowing the model to see the correct decision boundary) or C-Cifar10 (which still contains the bias, presenting a more challenging scenario). We evaluate the performance of both LP and k-NN on the U-Cifar10 validation set (see Fig. 2).

Table 1: Accuracy comparison between baselines and our approach with side information (-SI). The encoders are pre-trained on C-Cifar10: the LP/k-NN classifiers are either fitted with C-Cifar10 or U-Cifar10, and always validated on U-Cifar10 (spurious correlation removed).

| Method | $\gamma$ | LP: C→U | k-NN: C→U | LP: U→U | k-NN: U→U |
|---|---|---|---|---|---|
| Barlow Twins | – | 52.19 ± 0.65 | 45.22 ± 0.38 | 82.93 ± 0.40 | 71.56 ± 0.60 |
| Barlow Twins-N | – | 51.89 ± 0.68 | 44.56 ± 0.63 | 83.48 ± 0.17 | 71.96 ± 0.49 |
| Barlow Twins-SI | 1280 | **66.14 ± 0.83** | **62.82 ± 0.40** | **83.97 ± 0.35** | **78.33 ± 0.82** |
| CorInfoMax | – | 47.22 ± 0.30 | 36.65 ± 0.46 | 82.81 ± 0.16 | 71.31 ± 0.31 |
| CorInfoMax-N | – | 46.19 ± 0.23 | 35.58 ± 0.43 | 83.10 ± 0.41 | 71.17 ± 0.85 |
| CorInfoMax-SI | 20.0 | **60.29 ± 0.08** | **54.88 ± 0.43** | **83.69 ± 0.77** | **75.11 ± 0.42** |
| SepCLR | – | 58.00 ± 0.97 | 53.95 ± 1.36 | 81.33 ± 0.42 | 66.68 ± 0.74 |

**Barlow Twins and CorInfoMax.** We first evaluate our method using Barlow Twins (Zbontar et al., 2021), extending the original objective with our JSD term. The modified loss becomes:

$$\mathcal{L}_{BT-SI} = \sum_i (1 - \boldsymbol{C}_{i,i})^2 + \eta \sum_i \sum_{j \neq i} \boldsymbol{C}_{i,j}^2 - \gamma \, \mathrm{JSD}_\alpha(\bar{z}_{\mathcal{X}} \| \bar{z}_{\mathcal{S}}) \tag{5}$$

where $\boldsymbol{C}$ is the cross-correlation matrix between paired views, and $\bar{z}_{\mathcal{X}}$, $\bar{z}_{\mathcal{S}}$ denote the average softmax outputs over samples from the main and side datasets, respectively. To show that out approach is not tied to any specific method, we also apply it to CorInfoMax Ozsoy et al. (2022), an information-maximization-based JEA. Specifically, we augment the original loss with the proposed JSD term:

$$\mathcal{L}_{CIM-SI} = \eta \| \boldsymbol{Z}^{(1)} - \boldsymbol{Z}^{(2)} \|_F^2 - (\log |\boldsymbol{R}_{\boldsymbol{z}^{(1)}} + \epsilon \boldsymbol{I}| + \log |\boldsymbol{R}_{\boldsymbol{z}^{(2)}} + \epsilon \boldsymbol{I}|) - \gamma \, \mathrm{JSD}_\alpha(\bar{z}_{\mathcal{X}} \| \bar{z}_{\mathcal{S}}) \tag{6}$$

where $\boldsymbol{R}_z$ is the auto-covariance matrix for each view. The first term encourages alignment of different views, while the second encourages high information content in the representations. Table 1 shows results for different configurations. The most informative setting is when the classifier is trained on C-Cifar10 and tested on U-Cifar10, as this reveals whether the learned representations

themselves overcome the spurious correlation. Training on U-Cifar10, by contrast, gives the classifier direct access to the correct decision boundary, making the task easier. First, we observe that incorporating side information naively (-N) provides no noticeable gains over the respective standard baselines. However, when the methods are encouraged to separate main and side representations through the Jensen-Shannon divergence (-SI), their ability to focus on the target features improves considerably, with higher accuracy as a result. SepCLR outperforms standard and naive models, but performs worse than -SI models, despite making use of a dedicated encoder to model nuisance features. We further study the weight $\gamma$ for the JSD term in Table 2. As $\gamma$ increases, performance on the challenging C→U improves consistently, indicating that a stronger incentive to disentangle nuisance information yields more robust features. At large values, some over-regularization on the simpler U→U setting is observed, suggesting a trade-off between nuisance suppression and preserving within-domain variability.

Table 2: Performance for different $\gamma$ (controlling the strength of the JSD term) for the SI methods.

| Method | $\gamma$ | LP: C→U | k-NN: C→U | LP: U→U | k-NN: U→U |
|---|---|---|---|---|---|
| Barlow Twins-SI | 160 | $60.33 \pm 0.63$ | $54.14 \pm 0.62$ | $85.88 \pm 0.29$ | $77.44 \pm 0.75$ |
| Barlow Twins-SI | 320 | $63.93 \pm 0.94$ | $58.80 \pm 1.20$ | $\mathbf{86.34 \pm 0.23}$ | $79.00 \pm 0.25$ |
| Barlow Twins-SI | 640 | $66.11 \pm 0.36$ | $62.23 \pm 0.22$ | $85.68 \pm 0.24$ | $\mathbf{79.34 \pm 0.45}$ |
| Barlow Twins-SI | 1280 | $\mathbf{66.14 \pm 0.83}$ | $\mathbf{62.82 \pm 0.40}$ | $83.97 \pm 0.35$ | $78.33 \pm 0.82$ |
| CorInfoMax-SI | 1.0 | $48.17 \pm 0.58$ | $37.94 \pm 0.39$ | $83.72 \pm 0.64$ | $72.99 \pm 0.71$ |
| CorInfoMax-SI | 5.0 | $53.85 \pm 0.47$ | $45.06 \pm 0.41$ | $85.50 \pm 0.11$ | $76.34 \pm 0.31$ |
| CorInfoMax-SI | 10.0 | $57.97 \pm 0.85$ | $51.24 \pm 0.67$ | $\mathbf{85.75 \pm 0.38}$ | $\mathbf{77.56 \pm 0.17}$ |
| CorInfoMax-SI | 20.0 | $\mathbf{60.29 \pm 0.08}$ | $\mathbf{54.88 \pm 0.43}$ | $83.69 \pm 0.77$ | $75.11 \pm 0.42$ |

## 5.2 Application to Colonoscopy

To show the impact of leveraging side information on real-world applications, we evaluate our method on two clinically-relevant downstream tasks in colonoscopy video analysis: 1) *Polyp histology classification*: classifying hyperplastic vs adenomatous polyps, and 2) *Polyp morphology classification*: classifying the polyp's form and structure.

**Masked Siamese Networks.** We adapt the MSN framework (Assran et al., 2022) by incorporating our side information method. In addition to the original cross-entropy loss between anchor and target predictions $p^{(a)}$ and $p^{(t)}$, we compute the JSD between aggregated anchor and target predictions across main and side samples. The resulting objective is

$$\mathcal{L}_{\text{MSN-SI}} = \underbrace{\frac{1}{BM} \sum_{i=1}^{B} \sum_{j=1}^{M} H\left(p_i^{(t)}, p_{i,j}^{(a)}\right)}_{\text{cross-entropy}} - \lambda \underbrace{H\left(\bar{p}^{(a)}\right)}_{\text{ME-MAX}} - \gamma[\underbrace{\text{JSD}_\alpha\left(\bar{p}_{\mathcal{X}}^{(a)} \parallel \bar{p}_{\mathcal{S}}^{(t)}\right)}_{\text{anchor vs. side target}} + \underbrace{\text{JSD}\left(\bar{p}_{\mathcal{S}}^{(a)} \parallel \bar{p}_{\mathcal{X}}^{(t)}\right)}_{\text{side anchor vs. target}}]$$

**Colonoscopy Data.** For pre-training, we use REAL-Colon (Biffi et al., 2024), a large and public dataset with around $2.7M$ frames from 60 recordings. REAL-Colon provides full length colonoscopy screenings, meaning that a majority of these frames are negatives without any polyps. There are in total $\sim 350K$ bounding box annotations, defining the set of positive images. The rest of the dataset is considered as the side information. We use two downstream datasets. PolypsSet (Li et al., 2021) provides bounding box annotations and binary labels for adenoma and hyperplastic polyps, with $\sim 38K$ frames from 155 video sequences split on sequence level into $75\%, 10\%, 15\%$ train, validation, and test. The SUN database (Misawa et al., 2021) contains $\sim 49K$ frames taken from 100 different polyps with morphology labels. We split at the polyp level ($60\%/20\%/20\%$) with class-proportion stratification and binarize the morphology classes to create our task by grouping Is, Isp, and Ip into Class I and IIa and IIa (LST-NG) into Class II, following the Paris grouping (Lambert, 2003). For both the histology and morphology classification tasks we perform linear probing. We compare our results to those reported by Hirsch et al. (2023), noting that their models were pre-trained on different datasets—both public and private—than ours, which must be taken into account in the comparisons.

**Results.** We report macro F1 test results for the polyp histology classification task on PolypsSet in Table 3. A model pre-trained on REAL-Colon with our choice for hyper-parameters (without incorporating side information) outperforms the best previous models pre-trained on public data by 1.5%, and by 5.5% when comparing models with identical architectures, but underperforms when compared to models pre-trained on larger private datasets. The naive incorporation (MSN-N) improves the results by another 1.7%. However, when using our proposed method (MSN-SI), we achieve a F1 macro score of 80.3%, matching the best privately trained models *while using an order of magnitude less data and fewer parameters*. This demonstrates that, when informative data is limited but relevant side information is available, our method can learn useful features more efficiently — compensating for the data disadvantage through auxiliary structure. In Table 4 we see how the downstream performance changes when incorporating different ratios of side information. Across both tasks, MSN-SI outperforms the standard baseline (MSN) at every negative-ratio setting, and it surpasses MSN-N in almost all comparisons.

Table 3: F1 test performance on PolypsSet histology classification. Supervised learning (SL) and SSL pre-training on private and public datasets are compared. Note that data differs between our setting (bottom part) and that of Hirsch et al. (2023) (upper part), their private data being one order of magnitude bigger than our public. This shows that our method learns useful features more efficiently.

| Method | Framework | Arch | Private | Public |
|---|---|---|---|---|
| FS (Ramesh et al., 2023) | SL | RN50 | - | 72.1 |
| DINO (Ramesh et al., 2023) | SSL | RN50 | - | 72.4 |
| MSN (Hirsch et al., 2023) | SSL | ViT-S | 78.5 | 70.6 |
| MSN (Hirsch et al., 2023) | SSL | ViT-B | 78.2 | 74.6 |
| MSN (Hirsch et al., 2023) | SSL | ViT-L | **80.4** | 73.6 |
| MSN | SSL | ViT-S | - | 76.1 |
| MSN-N (ours) | SSL | ViT-S | - | 77.8 |
| MSN-SI (ours) | SSL | ViT-S | - | **80.3** |

Table 4: Average F1, Precision, and Recall for different negative ratios for histology (PolypsSet) and morphology (SUN) classification. Standard deviations obtained by training multiple linear probes.

| $R_{SI}$ | Method | PolypsSet | | | SUN | | |
|---|---|---|---|---|---|---|---|
| | | F1 | Precision | Recall | F1 | Precision | Recall |
| 0 | MSN | 76.1 ± 0.3 | 77.4 ± 0.2 | 75.4 ± 0.4 | 70.5 ± 0.6 | 76.0 ± 1.4 | 68.8 ± 0.5 |
| 12.5 | MSN-N | 75.9 ± 0.3 | 76.9 ± 0.2 | 75.4 ± 0.4 | 77.2 ± 0.5 | 82.8 ± 2.1 | 75.0 ± 0.6 |
| | MSN-SI | 77.5 ± 0.4 | 78.5 ± 0.2 | 76.9 ± 0.5 | 74.0 ± 1.0 | 79.5 ± 1.0 | 72.2 ± 1.2 |
| 25 | MSN-N | 77.2 ± 0.1 | 78.7 ± 0.2 | 76.5 ± 0.1 | 71.2 ± 1.1 | 79.3 ± 3.7 | 69.3 ± 0.8 |
| | MSN-SI | 80.3 ± 0.1 | 80.5 ± 0.1 | 80.1 ± 0.2 | 72.5 ± 0.4 | 78.4 ± 2.6 | 70.8 ± 0.8 |
| 50 | MSN-N | 77.8 ± 0.4 | 78.0 ± 0.3 | 77.6 ± 0.4 | 72.8 ± 1.6 | 83.7 ± 0.8 | 70.5 ± 1.5 |
| | MSN-SI | 78.0 ± 0.2 | 78.9 ± 0.2 | 77.5 ± 0.2 | 74.6 ± 1.0 | 83.5 ± 0.4 | 72.2 ± 1.0 |

## 5.3 SENSITIVITY AND HYPERPARAMETERS

NF–MV introduces two hyperparameters: the JSD weight $\gamma$ and the side-information ratio $R_{SI}$, which specifies the proportion of main versus side samples in each minibatch. This ratio directly determines the weighting parameter $\alpha$ used in the weighted JSD objective in eq. 4. Although simple, these two parameters govern the balance between (i) the standard MultiView alignment objective and (ii) the nuisance-separation signal provided by side information. We summarise their behaviour below and highlight consistent patterns that appear across all SSL backbones.

**JSD weight $\gamma$.** The coefficient $\gamma$ controls the relative scale of the JSD penalty with respect to the underlying SSL loss. Across all methods we evaluated (Tables 2, 7), we observe a broad stability region: small values introduce only a mild separation effect, while moderate values reliably improve robustness to nuisance correlations without requiring fine-tuning. As $\gamma$ increases further, the encoder allocates more of its capacity to identifying nuisance structure, which strengthens the main–side

contrast but may lead to over-regularization if the JSD term begins to dominate the optimisation dynamics.

**Side-information ratio** $R_{SI}$**.** The ratio $R_{SI}$ determines how frequently side samples appear within a minibatch and therefore how much gradient budget is allocated to modelling nuisance structure. This parameter plays a similar role to environment sampling in domain-adversarial or contrastive-analysis settings. Across experiments, we find that NF–MV is effective for a wide range of moderate ratios (Tables 4, 8, 9). Too little side information does not allow the model to learn sufficient representations of the nuisance, while a too high ratio drowns out the learning signal targeted for the main data. Importantly, these trends are stable across architectures (Barlow Twins, CorInfoMax, VICReg, BYOL), suggesting that $R_{SI}$ primarily controls the amount of side signal rather than interacting idiosyncratically with a particular SSL design.

Together, $\gamma$ and $R_{SI}$ form a simple and interpretable interface: $\gamma$ regulates how strongly nuisance structure is separated, while $R_{SI}$ determines how much such structure is observed during training. In practice, both parameters exhibit wide robustness regions, and we provide default settings in our code that reproduce the results reported in the paper.

**Side information quality.** Consistency with the NF–MV assumption requires that the side set be approximately task-irrelevant. To assess how deviations from this assumption affect performance, we introduce controlled contamination by injecting task-relevant signal into the side set (Table 10). As expected, performance decreases as the contamination level increases, since the contrast between the main and side distributions becomes weaker. However, degradation is gradual rather than abrupt: even with non-trivial contamination, NF-MV continues to outperform the baseline. This indicates that the method does not rely on perfectly curated side information and tolerates moderate violations of $I(Y; X_{\text{side}}) = 0$.

The behaviour is explained by the gradient structure of the JSD term (Appendix D.6). As the two domain-conditional feature distributions move closer, the population gradient of the JSD vanishes, while the minibatch estimator retains only finite-sample noise. Thus, contamination increases the noise-to-signal ratio of the JSD gradient but does not collapse the objective. This theoretical property aligns with the empirical results: NF–MV becomes less effective under heavy contamination, yet remains stable and beneficial under moderate impurity levels.

## 6 CONCLUSION

Self-supervised learning (SSL) has advanced significantly, often leveraging the assumption that different views of the same input contain task-relevant information. However, we revisited this foundational *MultiView assumption* and showed that it can be overly permissive in practical settings—particularly when nuisance factors such as background textures or procedural artifacts persist across views. These shared but task-irrelevant features can entangle with useful signals and degrade downstream performance. To address this limitation, we introduced the *Nuisance-Free MultiView* (NF-MV) assumption, which formally distinguishes between shared, relevant information and persistent nuisance structure. Building on this perspective, we proposed a general and architecture-agnostic framework for incorporating *side information* into joint embedding pre-training. This enables learning representations that are sufficient for the task while being invariant to nuisance factors. Our method integrates a simple Jensen–Shannon divergence term into the SSL objective, penalizing representational overlap between main and side data. This simple approach proves effective across both controlled synthetic setups and complex real-world domains like colonoscopy video analysis. Crucially, the kind of side information we exploit is often naturally present in real-world data pipelines but routinely discarded during dataset curation or ignored during training. Our results show that such data, when used appropriately, can serve as a powerful signal for guiding representation learning—not by telling models what to learn, but by clarifying what not to learn. This shift in perspective has the potential to improve generalization when task-irrelevant structure is abundant.

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

## A  JENSEN-SHANNON DIVERGENCE AND MUTUAL INFORMATION

While the connection between mutual information and the Jensen–Shannon Divergence is well-known for the equiprobable setting, here we prove a more general relationship in the non-equiprobable setting using the $\alpha$-weighted Jensen–Shannon divergence. The Jensen-Shannon Divergence (JSD) is a symmetrized version of the Kullback-Leibler divergence KL.

**Definition 2** (Jensen-Shannon Divergence). *Let $P, Q$ be two distributions, and $M$ the mixture $\frac{1}{2}(P + Q)$. Then:*

$$\mathrm{JSD}(P\|Q) = \frac{1}{2}\,\mathrm{KL}(P\|M) + \frac{1}{2}\,\mathrm{KL}(Q\|M), \tag{7}$$

It is well known that the mutual information between a random variable $Z$ associated to the mixture $M = \frac{1}{2}(P + Q)$ and the (equiprobable) binary indicator $B$ – that specifies whether $Z$ was drawn from $P$ or $Q$ – can be expressed as $\mathrm{JSD}(P\|Q)$:

$$\begin{aligned}
I(Z;B) &= H(B) - H(B|Z) \\
&= -\sum M \log M + \frac{1}{2}\left(\sum P \log P + \sum Q \log Q\right) \\
&= -\sum \frac{P}{2} \log M - \sum \frac{Q}{2} \log M + \frac{1}{2}\left(\sum P \log P + \sum Q \log Q\right) \\
&= \frac{1}{2}\sum P \log \frac{P}{M} + \frac{1}{2}\sum Q \log \frac{Q}{M} \\
&= \mathrm{JSD}(P\|Q)
\end{aligned}$$

The above assumes the mixture is even, and so that the binary indicator is equiprobable with $\mathbb{P}(B = 0) = \mathbb{P}(B = 1) = \frac{1}{2}$. This can be extended to uneven mixtures, $M_\alpha = \alpha P + (1-\alpha)Q$. In this setting the indicator is not equiprobable, instead $\mathbb{P}(B = 0) = \alpha$ and the standard JSD loses it connection between the indicator and mutual information. However, it can be recovered by considering a family of weighted Jensen-Shannon divergences.

**Definition 3** (Weighted Jensen-Shannon Divergence, (Nielsen, 2020)). *Let $P, Q$ be two distributions, and let $M_\omega = \omega P + (1 - \omega)Q$. Then:*

$$\mathrm{JSD}_\omega(P\|Q) = \omega\,\mathrm{KL}(P\|M_\omega) + (1 - \omega)\,\mathrm{KL}(Q\|M_\omega). \tag{8}$$

With this definition it is possible to extend the above result to a more general setting. Let the weight in $\mathrm{JSD}_\omega$ be equal to $\alpha = \mathbb{P}(B = 0)$. Let $Z$ be a random variable associated with the mixture $M_\alpha = \alpha P + (1 - \alpha)Q$, so that $\mathbb{P}(B = 0) = \alpha$ and $\mathbb{P}(B = 1) = (1 - \alpha)$. Then:

$$\begin{aligned}
I(Z;B) &= H(B) - H(B|Z) \\
&= -\sum M_\alpha \log M_\alpha + \left(\alpha \sum P \log P + (1 - \alpha)\sum Q \log Q\right) \\
&= -\alpha \sum P \log M_\alpha - (1 - \alpha)\sum Q \log M_\alpha + \left(\alpha \sum P \log P + (1 - \alpha)\sum Q \log Q\right) \\
&= \alpha \sum P \log \frac{P}{M_\alpha} + (1 - \alpha)\sum Q \log \frac{Q}{M_\alpha} \\
&= \alpha\,\mathrm{KL}_\alpha(P\|M_\alpha) + (1 - \alpha)\,\mathrm{KL}(Q\|M_\alpha) \\
&= \mathrm{JSD}_\alpha(P\|Q)
\end{aligned}$$

We state this as a lemma:

**Lemma 1.** *Let $P, Q$ be two distributions and consider the mixture distribution $M_\alpha = \alpha P + (1 - \alpha)Q$. Define the binary variable $B$ indicating from which distribution $Z$ was drawn, such that $\mathbb{P}(B = 0) = \alpha$ and $\mathbb{P}(B = 1) = (1 - \alpha)$. Then the mutual information between $Z$ and the indicator $B$ is the weighted Jensen-Shannon divergence, with weight $\alpha$:*

$$I(Z;B) = \mathrm{JSD}_\alpha(P\|Q).$$

# B  MINIMAL REPRESENTATION AND GENERALIZATION ERROR

## B.1  TECHNICAL PRELIMINARIES

**Mutual information and entropy.**  For random variables $U, V$ on finite or countable alphabets,

$$I(U;V) = H(U) - H(U|V) = H(V) - H(V|U).$$

Key properties exploited in the proof are:

\* **Data-processing inequality:** if $U \to V \to W$, then $I(U;W) \leq I(U;V)$.

\* **Entropy upper-bounds mutual information:** $I(U;V) \leq H(U)$ by non-negativity of entropy.

\* **Sub-additivity of entropy:** $H(U_1, \ldots, U_n) \leq \sum_{i=1}^{n} H(U_i)$.

**Sub-Gaussian random variables.**  A zero-mean random variable $Z$ is called $\sigma$-*sub-Gaussian* if $\mathbb{E}[\exp(\lambda Z)] \leq \exp(\lambda^2 \sigma^2 / 2)$ for all $\lambda \in \mathbb{R}$. The sub-Gaussian condition ensures that the empirical-to-population loss difference concentrates at a $\sqrt{1/n}$ rate, which underpins the Xu–Raginsky bound below.

**Xu–Raginsky generalization bound.**  For a fixed learning algorithm $\mathcal{A}$ and any sample size $n$,

$$\left| \mathrm{gen}(W, S) \right| := \left| \mathbb{E}\left[ \ell(W, (X, Y)) \right] - \frac{1}{n} \sum_{i=1}^{n} \ell(W, (X_i, Y_i)) \right| \leq \sqrt{\frac{2\sigma^2}{n} I(S; W)}.$$

## B.2  THEOREM AND PROOF

The following theorem is an adaptation of the Xu & Raginsky bound (Xu & Raginsky, 2017, Thm. 1). We restate the theorem from the main paper, Theorem 1.

**Theorem 1** (Generalization Benefit of Task-Specific Minimality). *Let $\mathcal{T}$ be a supervised learning task with distinct alphabet $\mathcal{Y}$ and let $\mathcal{T}' \subset \mathcal{T}$ be a strict sub-task. Let $Z = f(X)$ be minimal sufficient for $\mathcal{T}$ and $Z' = f'(X)$ be minimal sufficient for $\mathcal{T}'$. Draw a training set $S = (X_1, \ldots, X_n) \sim \mathcal{D}^n$, and let a fixed learning algorithm yield hypothesises $W = \mathcal{A}(Z^n)$ and $W' = \mathcal{A}((Z')^n)$. Suppose the loss $\ell(W, (X, \mathcal{T}'))$ is $\sigma$-sub-Gaussian. Then*

$$\mathbb{E}_{S,W}\left[ \mathrm{gen}(W, S) \right] \leq \sigma \sqrt{2 I(Z; X)}, \qquad \mathbb{E}_{S,W'}\left[ \mathrm{gen}(W', S) \right] \leq \sigma \sqrt{2 I(Z'; X)},$$

*so the upper bound for the generalisation error for $Z'$ is strictly tighter, as $I(Z'; X) < I(Z; X)$.*

*Proof. Information ordering.* Because $\mathcal{T}' \subset \mathcal{T}$, any encoder sufficient for $\mathcal{T}$ is sufficient for $\mathcal{T}'$, so minimality gives $I(Z'; X) \leq I(Z; X)$. If equality held, $Z$ would also be minimal for $\mathcal{T}'$, contradicting the assumption that the tasks are distinct. Hence $I(Z'; X) < I(Z; X)$.

**Xu–Raginsky bounds.**  Xu and Raginsky (Xu & Raginsky, 2017, Thm. 1) give, for any training set $S$ and hypothesis $W$,

$$\left| \mathrm{gen}(W, S) \right| \leq \sqrt{\frac{2\sigma^2}{n} I(S; W)}.$$

We now upper-bound $I(S; W)$ by $n I(Z; X)$ in four steps.

i **Data-processing.** $Z^n = f(S)$ with $f$ deterministic and fixed, hence

$$I(S; W) \leq I(Z^n; W).$$

ii **Replace mutual information by entropy.** For any pair of r.v.'s $U, V$, $I(U; V) \leq H(U)$, so

$$I(Z^n; W) \leq H(Z^n).$$

iii **Sub-additivity of entropy.** Entropy is sub-additive, $H(Z^n) \leq \sum_{i=1}^{n} H(Z_i)$.

iv **Deterministic encoder.** Because each $Z_i = f(X_i)$ is a deterministic function of $X_i$, we have $H(Z_i \mid X_i) = 0$ and therefore

$$H(Z_i) \;=\; I(Z_i; X_i) \;=\; I(Z; X).$$

Summing over $i$ yields $\sum_{i=1}^{n} H(Z_i) = n\, I(Z; X)$.

Combining (i)–(iv) gives the desired bound

$$I(S; W) \;\leq\; n\, I(Z; X),$$

so that

$$\mathbb{E}_{S,W}\big[\mathrm{gen}(W, S)\big] \;\leq\; \sigma\sqrt{2\, I(Z; X)}.$$

Applying the same four-step argument with $Z'$ in place of $Z$ produces the second inequality with $I(Z'; X)$. Because $I(Z'; X) < I(Z; X)$, the bound for $Z'$ is strictly tighter.

*Tight-bound case.* When $\mathcal{A}$ saturates the Xu–Raginsky bound, the ordering of bounds becomes the ordering of the expected generalization errors. $\qquad\square$

### B.3 DISCUSSION

Intuitively, the less information an encoder retains about the raw input $X$, the fewer spurious correlations can be memorised by a learning algorithm $\mathcal{A}$, and the harder it becomes to over-fit finite samples.

The statement formalises this intuition by comparing the information–risk trade-off of two encoders:

$Z = f(X)$ is *minimal sufficient* for the *parent task* $\mathcal{T}$; $Z' = f'(X)$ is minimal sufficient for the *sub-task* $\mathcal{T}'$, with $\mathcal{T}' \subset \mathcal{T}$.

Because every predictor that solves $\mathcal{T}$ necessarily solves the smaller task, a representation that is minimal for $\mathcal{T}'$ *cannot contain more* information about $X$ than one that is minimal for $\mathcal{T}$. The strict inclusion $\mathcal{T}' \subset \mathcal{T}$ makes this comparison *strict*, leading to the inequality $I(Z'; X) < I(Z; X)$.[4]

**Assumptions.** The result rests on two assumptions that deserve emphasis.

1. **Deterministic encoders.** The proof bounds $H(Z^n)$ via $H(Z_i) = I(Z; X)$, which uses $H(Z_i|X_i) = 0$.

2. $\sigma$**-sub-Gaussian loss.** The Xu–Raginsky inequality applies only when the per-sample loss is sub-Gaussian; heavy-tailed losses need alternative concentration tools.

## C LIMITATIONS

*Side information availability.* Our method assumes access to auxiliary data capturing task-irrelevant structure (e.g., the $\sim 87\%$ polyp-negative frames in REAL-Colon). While such side information is often available in practice—naturally collected by endoscopes, cameras, and sensors—it is typically discarded during dataset curation in favour of compact, label-dense benchmarks. *MI proxy*. We use the Jensen-Shannon divergence between empirical feature distributions as a tractable proxy to penalize representational overlap with side information. However, this measure is coarse and may not fully capture the underlying interactions. Future work may explore alternatives such as contrastive bounds, adversarial losses, or kernel-based dependence measures. *Nuisance–task independence*. Treating a dataset as side information assumes it contains only nuisance features. If the side data includes task-relevant signals, this assumption is violated and performance may degrade - however the proposed method is relatively robust with respect to this (see Table 10).

---

[4] A typical example is image classification: a representation sufficient for recognising *all* ImageNet classes carries more bits about the input than one sufficient only for, say, the binary "cat–versus–not-cat" sub-task.

# D  ABLATIONS AND IMPLEMENTATIONS FOR CIFAR10+MNIST EXPERIMENTS

We specify the hyper-parameters and settings for the Cifar10+MNIST experiments here. A visualisation of the setting (pre-training on correlated data, probing on correlated/uncorrelated data, testing on uncorrelated data) can be seen in Fig. 2.

**Augmentations.**  During pre-training we use the transformations defined in Table 5. We also normalize the data with per-channel mean and standard deviation.

Table 5: Augmentations used during pre-training of CorInfoMax methods. Barlow Twins use the same transformations, with the exception of Gaussian blur which is not used. $A_1$ and $A_2$ are used to create the two different views of the same image. RRC denotes random resized crop and CJ denotes colour jitter.

| **Transformation** | $A_1$ | $A_2$ |
|---|---|---|
| RRC-prob. | 1.0 | 1.0 |
| RRC-scale | [0.08, 1] | [0.08, 1] |
| RRC-size | 32 | 32 |
| CJ-prob. | 0.8 | 0.8 |
| CJ-brightness offset | 0.4 | 0.4 |
| CJ-Contrast offset | 0.4 | 0.4 |
| CJ-Saturation offset | 0.2 | 0.2 |
| CJ-Hue max | 0.1 | 0.1 |
| Horizontal flip prob. | 0.5 | 0.5 |
| Grayscale prob. | 0.2 | 0.2 |
| Gaussian blur prob. | 1.0 | 0.1 |
| Solarization | 0.0 | 0.2 |

**Evaluation.**  We evaluate learned representations using two methods: a *linear classifier* and a *weighted k-NN classifier*. For linear probing, we train a linear classifier on frozen features for 100 epochs using SGD with momentum 0.9 and no weight decay. The learning rate follows a cosine decay schedule, starting at 0.2 and decaying to a minimum of 0.002. During training, we apply only random horizontal flipping (probability 0.5) and normalization; no augmentations are applied to the validation set aside from normalization using training-set statistics. For the k-NN evaluation, we use a weighted k-NN classifier with temperature $T = 0.5$ and $k = 200$ neighbours. In all cases, we train the probe/k-NN on either C-Cifar10 or U-Cifar10, and evaluate their performance on the uncorrelated (U-Cifar10) validation set. When probing using U-Cifar10 we allow the model to see data without the correlation, and unlearn the shortcut. The most challenging case is when the probing data also contains the bias, using C-Cifar10 both for training the classifier and the encoder.

## D.1  CORINFOMAX

Our implementation and hyper-parameter selection is based on the original implementation of CorInfoMax from Ozsoy et al. (2022). Our implementation was also tested on regular CIFAR10, to assert correctness, showing performance that aligns with the original implementation. All hyper-parameters are chosen based on single runs to keep the number of experiments feasible.

**Architecture and Projector.**  As is standard we use a modified ResNet-18 without max pooling and a $3 \times 3$ kernel for the first convolutional layer to accommodate for the low resolution images. The projection head is a 3-layer MLP $[2048 - 2048 - 64]$.

**Optimization.**  All models are pre-trained for 1000 epochs with a batch size of 512 using SGD (momentum 0.9, weight decay $1e - 4$). The learning rate follows a cosine decay schedule with linear warm-up. The starting learning rate is $0.003$, which increases over 10 warm-up epochs to the maximum learning rate of $0.5$. The minimum learning rate is set at $1e - 6$.

**Loss scale.** In the original implementation it is reported that using $\eta = 250$ yields the best performance after having tried $\eta \in [250, 500, 1000]$. In our experiments we performed a sweep over $\eta \in [100, 250]$ for the baselines. We find that using $\eta = 100$ performs best in our setting (see Table 6). We hypothesise that this is due to our choice for the pre-training dataset that now carries less information about the downstream task as compared with the original Cifar10 setting. Using the optimal hyper-parameters found for the baselines, we conducted a sweep over $\gamma \in [1, 5, 10, 20]$ to find the appropriate weighting for the additional loss term associated with side information. We use warm-up and a linear schedule for $\gamma$, to allow the model to learn stable representations of the side information before removing it as discussed in Section 4. The warm-up lasts for 100 epochs with $\gamma_w = 0$, which then linearly increases for the remaining of the training to the final value $\gamma$.

Table 6: Accuracy for the baselines CorInfoMax and CorInfoMax-N over different values of $\eta$.

| Method | $\eta$ | $R_{SI}$ | LP: C→U | k-NN: C→U | LP: U→U | k-NN: U→U |
|---|---|---|---|---|---|---|
| CorInfoMax | 100 | - | 47.26 | 36.51 | 82.94 | 71.57 |
| CorInfoMax | 250 | - | 45.78 | 33.65 | 82.27 | 70.24 |
| CorInfoMax-N | 100 | 10% | 46.04 | 35.18 | 83.35 | 70.88 |
| CorInfoMax-N | 250 | 10% | 45.89 | 33.76 | 83.57 | 71.38 |

## D.2 BARLOW TWINS

Our implementation is based on da Costa et al. (2022), since this achieves better performance than the original implementation of Barlow Twins Zbontar et al. (2021). Our implementation was also tested on regular CIFAR10, to assert correctness, showing performance that aligns with the implementation from da Costa et al. (2022).

**Architecture and Projector.** We use a ResNet-18 without max pooling and a $3 \times 3$ kernel for the first convolutional layer to accommodate for the low resolution images. The projection head is a 3-layer MLP $[2048 - 2048 - 2048]$.

**Optimization.** All models are pre-trained for 1000 epochs with a batch size of 256 using LARS You et al. (2017) (trust coefficient 0.2, weight decay $1e - 4$, exclude bias and norm True). The learning rate follows a cosine decay schedule with linear warm-up. The starting learning rate is $3e - 5$, which increases over 10 warm-up epochs to the maximum learning rate of 0.3. Minimum learning rate is set to 0.

**Loss scale.** The invariance weight $\eta$ is set to 0.0051, and the total loss scaled with 0.1. We do not apply the loss scaling to our additional loss term. We perform a sweep $\gamma \in [160, 320, 640, 1280]$ to see its effect. As with CorInfoMax-SI we use warm-up and a linear schedule for $\gamma$, to allow the model to learn stable representations of the side information before introducing the weighted Jensen-Shannon divergence loss. The warm-up lasts for 100 epochs with $\gamma_w = 0$, which then linearly increases for the remaining of the training to the final value $\gamma$.

## D.3 SEPCLR

Our implementation of SepCLR direcly follows that of the original from Louiset et al. (2024b).

**Architecture and Projector.** We use a ResNet-18 without max pooling and a $3 \times 3$ kernel for the first convolutional layer to accommodate for the low resolution images for both the salient and common encoder. The projection heads are 3-layer MLPs $[32 - 128 - 32]$.

**Optimization.** The models are pretrained for 500 epochs with a batch-size of 512 using the Adam optimizer (Kingma & Ba, 2017). While Barlow Twins and CorInfoMax was pretrained for 1000 epochs, training SepCLR for 500 epochs consumes about the same computational efforts (slightly more), and the training saturated. Following their implementation a constant learning rate of $3e - 4$ is used, with momentum 0.9 and no weight-decay.

**Augmentations.** In the reported experiments, the model was trained with augmentations as described in the original paper, with the only difference being the crop-size used, as we used 32x32 sized crops during training and evaluation to conform with the other experiments. We also tried using stronger augmentations, as used in our Barlow Twins experiments, but did not see any improvements from this.

**On the Comparison.** SepCLR trains one salient encoder and one target encoder. The goal of the salient encoder is to learn the digit representations, which we are interested in. Thus, the common encoder is not used for downstream testing in our experiments. Furthermore, the salient encoder learns a euclidean feature space, different from what is commonly used in SSL, where $\ell_2$ normalised features are most often used. Due to this, we evaluate the salient encoder without normalising the raw features from the salient backbone, as we find this improves the performance of SepCLR. For Barlow Twins and CorInfoMax we use normalisation. It should be noted that one of the strengths of SepCLR is to remove the salient variations from the common space - something we do not test for here, as this is not purpose of our work, and is difficult to compare between single and double encoder frameworks.

## D.4 ADDITIONAL RESULTS WITH VICREG AND BYOL

To further validate the approach, we have implemented our method for VICReg Bardes et al. (2022) and BYOL Grill et al. (2020). Both implementation follows that from da Costa et al. (2022), and the results can be seen in Table 7, where we see significant improvements for these methods as well, and similar hyperparameter patterns.

Table 7: Accuracy comparison between baselines and our approach with side information (-SI). The encoders are pre-trained on C-Cifar10: the LP/k-NN classifiers are either fitted with C-Cifar10 or U-Cifar10, and always validated on U-Cifar10 (spurious correlation removed).

| Method | $\gamma$ | LP: C$\rightarrow$U | k-NN: C$\rightarrow$U | LP: U$\rightarrow$U | k-NN: U$\rightarrow$U |
|---|---|---|---|---|---|
| VICReg | – | 49.64 | 44.82 | 79.20 | 64.00 |
| VICReg-SI | 40 | 55.11 | 50.44 | 82.38 | 69.50 |
| VICReg-SI | 80 | 59.94 | 55.03 | 84.27 | 73.08 |
| VICReg-SI | 160 | 65.65 | 61.48 | **84.88** | 76.06 |
| VICReg-SI | 320 | **66.55** | **62.96** | 83.48 | **76.60** |
| BYOL | – | 53.46 | 43.05 | 83.28 | 74.23 |
| BYOL-SI | 2 | 58.05 | 49.84 | **84.35** | 77.78 |
| BYOL-SI | 4 | 57.76 | 50.80 | 83.89 | **77.84** |
| BYOL-SI | 8 | **58.90** | **53.04** | 83.12 | 77.54 |

## D.5 SIDE INFORMATION RATIO

We investigate how the amount of side information in each batch affects the performance of the models. Here we have chosen the best performing hyper-parameters from Table 2 ($\gamma = 640$). We notice that combining a high $\gamma$ with a high ratio of side information $R_{SI}$ can destabilize the loss during training, leading to sub-optimal performance as seen in Tables 8 and 9.

Table 8: Accuracy over different negative ratios using Barlow Twins with side information.

| Method | $\gamma$ | $R_{SI}$ | LP: C$\rightarrow$U | k-NN: C$\rightarrow$U | LP: U$\rightarrow$U | k-NN: U$\rightarrow$U |
|---|---|---|---|---|---|---|
| BT | - | - | 51.50 | 44.98 | 83.05 | 70.86 |
| BT-SI | 640 | 12.5% | 66.44 | 62.61 | 85.60 | 79.19 |
| BT-SI | 640 | 25.0% | 67.83 | 64.99 | 85.52 | 78.72 |
| BT-SI | 640 | 50.0% | 61.04 | 57.41 | 81.07 | 74.31 |

Table 9: Accuracy over different negative ratios using CorInfoMax with side information.

| Method | $R_{SI}$ | LP: C→U | k-NN: C→U | LP: U→U | k-NN: U→U |
|---|---|---|---|---|---|
| CorInfoMax | - | 46.90 | 36.28 | 82.64 | 70.96 |
| CorInfoMax-N | 12.5% | 46.81 | 36.59 | 83.40 | 71.38 |
| CorInfoMax-N | 25.0% | 45.86 | 35.46 | 82.29 | 70.47 |
| CorInfoMax-N | 50.0% | 45.77 | 37.03 | 81.90 | 70.02 |
| CorInfoMax-SI | 12.5% | 60.90 | 54.97 | 83.07 | 75.07 |
| CorInfoMax-SI | 25.0% | 62.58 | 57.33 | 82.49 | 74.59 |
| CorInfoMax-SI | 50.0% | 55.36 | 51.64 | 75.66 | 66.82 |

## D.6 SENSITIVITY TO SIDE INFORMATION IMPURITY

To evaluate the impact of contamination of side data (where task relevant signals exist in the side information) on or method, we conduct empirical and qualitative analysis. In the empirical study, task-relevant information is introduced to the side data at different levels. For the qualitative we describe why the proposed JSD method is relatively roust to such imperfections.

**Empirical Study** A fraction of the side information is corrupted by replacing it with samples that contain the target feature, i.e. an MNIST digit. To preserve the correlated structure of the main setup, we used CIFAR10 as side information rather than CIFAR100, allowing for a one-to-one mapping between CIFAR10 classes and MNIST digits, while leaving all other settings unchanged. We then trained Barlow Twins encoders under different corruption ratios and evaluated transfer performance on U-CIFAR10 via linear probing and k-NN (Table 10). As expected, increasing the proportion of task-relevant side information consistently degrades performance, highlighting that the benefit of side information arises from its independence with respect to the main task. Yet, the model still outperforms the baseline, showing robustness to limited target features in the side information.

Table 10: Ablation on the effect of corrupting the side information with task-relevant signal. We replace a fraction of side data with CIFAR10–MNIST correlated pairs, while keeping all other settings unchanged. Performance is reported as linear probing (LP) and k-NN transfer from correlated to uncorrelated CIFAR10.

| Method | LP: C→U | k-NN: C→U |
|---|---|---|
| BT-SI  (0%) | 66.11 | 60.81 |
| BT-SI  (5%) | 63.84 | 58.41 |
| BT-SI  (20%) | 64.18 | 57.75 |
| BT-SI  (40%) | 61.37 | 55.23 |

**Qualitative Analysis** We begin by considering the behaviour of the JSD penalty in a limiting case of contamination. The JSD penalty promotes separation between $p_\theta(z \mid X_{\mathrm{main}})$ and $p_\theta(z \mid X_{\mathrm{side}})$; as contamination grows, these two feature distributions move closer. In the limiting case where $X_{\mathrm{main}}$ and $X_{\mathrm{side}}$ are drawn from the same input distribution, we have $p_\theta(z \mid X_{\mathrm{main}}) = p_\theta(z \mid X_{\mathrm{side}})$ for all $\theta$, and hence

$$\mathrm{JSD}\big(p_\theta(z \mid X_{\mathrm{main}}) \,\|\, p_\theta(z \mid X_{\mathrm{side}})\big) \equiv 0,$$

so its population gradient vanishes, $\nabla_\theta \mathrm{JSD} = 0$. In practice we optimise a minibatch estimator, whose expected gradient is then zero and whose residual contribution is due to finite-sample noise. This explains why the operational method (JSD) is robust to contamination of side information: it introduces additional noise into the gradients, but does not collapse the objective.

The analysis can be extended to the partial contamination case where $P_\theta := p_\theta(z \mid X_{\mathrm{main}})$ and $Q_\theta := p_\theta(z \mid X_{\mathrm{side}})$ become close. In this regime the log-density ratios $\log(P_\theta/M_\theta)$ and $\log(Q_\theta/M_\theta)$ shrink towards zero, and since JSD is an $f$-divergence, its population gradient vanishes at least quadratically in the distributional difference. A minibatch estimator, however, replaces the

expectations by averages over a fixed batch size $B$, so its variability is determined by finite-sample fluctuations of these log-ratio terms. These fluctuations decrease only through the usual $1/\sqrt{B}$ scaling of sample means and therefore do not vanish at the same rate as the population gradient when $P_\theta \to Q_\theta$. As a result, the *noise-to-signal ratio* of the JSD gradient increases as contamination grows; in the limit $P_\theta = Q_\theta$, the signal disappears while the estimator reduces to pure sampling noise. This explains why under contamination the JSD term becomes increasingly noisy, possibly hurting training dynamics.

This qualitative picture is consistent with our empirical study (Table 10), where performance degrades as side information becomes more contaminated, rather than collapsing learning.

### D.7 VISUALIZATION OF THE LEARNED REPRESENTATIONS

In Fig. 3 we show the result of applying t-SNE to visualise the raw features from the pre-trained backbones for Barlow Twins with and without side information. It is clear that both models perform better on the correlated validation set (C-Cifar10), where the Cifar10 shortcut can be leveraged. When the shortcut is removed (U-Cifar10), baseline Barlow Twins does not separate classes well. Instead, our proposed method (Barlow Twins with side information) can separate the classes even in this scenario, indicating that the correct discriminating features have been learned more effectively.

Figure 3: **Barlow Twins on (C,U)-Cifar10**: t-SNE visualizations of feature embeddings for the correlated (C-Cifar10, left) and uncorrelated (U-Cifar10, right) validation sets. Each row shows the learned representations from a different method, Barlow Twins (top) and Barlow Twins-SI (bottom). The colours represent the different MNIST classes in the (C,U)-Cifar10 images.

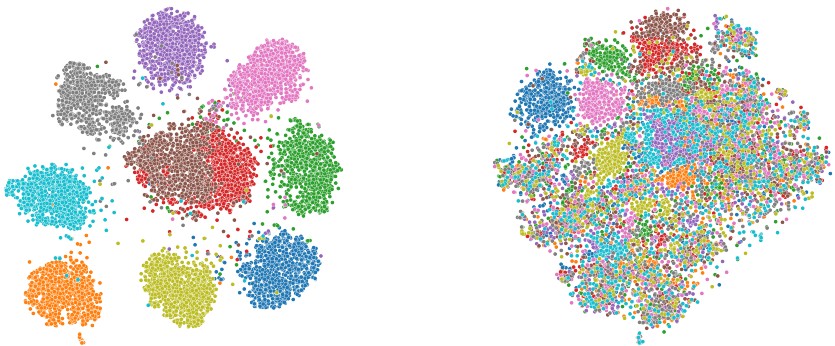

**Barlow Twins (Baseline):** Strong class separation on the correlated validation set (left), but major collapse on the uncorrelated set (right).

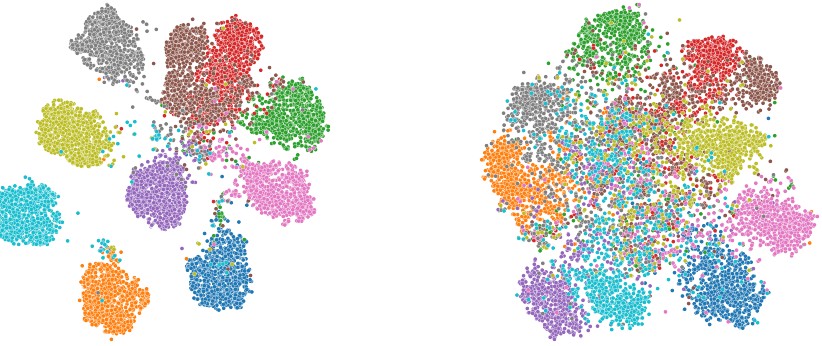

**Barlow Twins-SI (Ours):** Learns well-separated features on both domains, demonstrating better generalization to the uncorrelated setting.

# E  COLONOSCOPY EXPERIMENTS

## E.1  DATA PROCESSING AND SIDE INFORMATION

To obtain side information, we use the bounding box annotations provided by the REAL-Colon dataset. These annotations are precise, and some bounding boxes are very small. We therefore apply size-based filtering: bounding boxes smaller than 10% of the image diagonal or 10% of the image area are excluded during training. When sampling negative examples (used as side information), we sample uniformly across the entire dataset after subtracting the bounding boxes. If an image contains a bounding box, we extract a crop from outside the bounding box (as large as possible), applying the same thresholding criteria used for positive crops.

**Hyper-parameters.**  Significant computational resources and manual effort were devoted to tuning a strong baseline. All optimization hyper-parameters were selected based on performance on the validation split of the PolypsSet dataset, using a baseline model trained without side information. These hyper-parameters were then held constant across all model variants to ensure a fair comparison. We use a ViT-S backbone initialized from a DINO (Caron et al., 2021) checkpoint and pre-train for 30 epochs using the AdamW optimizer with a cosine learning rate schedule, including 5 warm-up epochs. A complete summary of the hyper-parameters is provided in Table 11.

Table 11: Pre-training hyper-parameters.

| Parameter | Value | Parameter | Value |
|---|---|---|---|
| Learning rate (start / final) | 0.0004 / 0.001 | Final tail LR | 0.001 |
| Weight decay | 0.01 | Clip gradient | 3.0 |
| Epochs | 30 | Cosine schedule | Yes |
| Warmup epochs | 5 | Batch size | 512 |
| Model | ViT-Small | Hidden dim | 2048 |
| Output dim | 256 | Drop path rate | 0.0 |
| Use BN / FP16 | True / False | Pretrained weights | DINO-ViT-S |

**Loss Configuration and SSL-Specific Parameters.**  We used ME-MAX regularization and Sinkhorn normalization. We found that using stronger ME-MAX regularization was beneficial in some settings, so we trained models with ME-MAX strength 1 and 3. Table 12 summarizes these settings.

Table 12: SSL loss configuration and architectural settings.

| Parameter | Value | Parameter | Value |
|---|---|---|---|
| ME-MAX regularization | Enabled | ME-MAX weight | 1.0 / 3.0 |
| Sinkhorn normalization | Enabled | Num. prototypes | 1024 |
| Temperature | 0.1 | Use sharpening | Yes |
| Sharpening start / final | 0.25 / 0.25 | Use Sinkhorn | True |

**Augmentations.**  We follow the augmentation pipeline proposed by Hirsch et al. Hirsch et al. (2023). Each image is first resized to $256 \times 256$ and then augmented into one global view and six focal views using a multi-crop strategy. Global views use a crop scale of $[0.5, 1.0]$ while focal views use $[0.1, 0.5]$. All views are normalized using domain-specific statistics. Colour jitter, grayscale augmentation, and horizontal flipping are applied stochastically. Table 13 lists the relevant parameters.

**Model Selection.**  For each method (MSN, MSN-N, MSN-SI), we selected the configuration that achieved the best performance on the PolypsSet validation set. Linear probing follows the procedure of Hirsch et al. (2023), with the difference that we use a single optimizer (Adam Kingma & Ba (2017)) throughout. The best-performing classifier checkpoint is selected from a single seed run,

Table 13: Data augmentation parameters.

| Parameter | Value | Parameter | Value |
|---|---|---|---|
| Image resize | (256, 256) | Color jitter strength | 0.5 |
| Global crop size | 224 | Focal crop size | 96 |
| Global crop scale | [0.5, 1.0] | Focal crop scale | [0.1, 0.5] |
| Rand / Focal views | 1 / 6 | Normalize mean | (0.656, 0.370, 0.268) |
| Normalize std | (0.0838, 0.0852, 0.0719) | Interpolation | Bicubic |
| Random flip | $p = 0.5$ | Grayscale | $p = 0.2$ |
| Masking Ratio | 75% | | |

after which we re-train the linear probes using multiple seeds to assess performance variability. Linear probes are trained for 20 epochs, as training typically plateaus early. Learning rates were selected via grid search over {1e-3, 5e-4, 1e-4}.

### E.2 SCHEMATIC OVERVIEW

A schematic view of how side information is incorporated in the MSN architecture can be seen in Fig. 4.

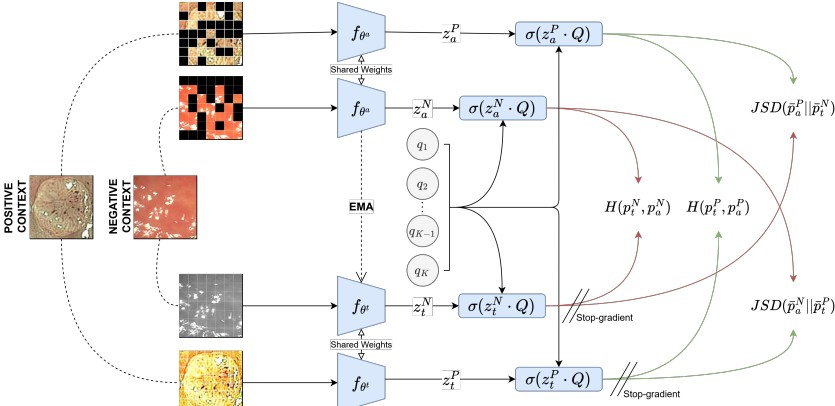

Figure 4: Network schematic for our MSN framework leveraging side information (MSN-SI).

**Compute and Memory.** The dataset used in our experiments, REAL-Colon, is large-scale, consisting of approximately 2.7 million high-resolution images occupying around 1TB of storage. For training, we use image crops defined by bounding boxes, resulting in roughly 350,000 samples. After applying a size threshold to filter out the smallest crops, this is reduced to about 300,000 images. When incorporating side information, the number of samples per epoch increases proportionally to the ratio of negative examples, which correspondingly raises the computational load. We apply a high masking ratio of 75%, which helps reduce computational demands. Nevertheless, due to the large dataset size and the generation of multiple crops per image using the MultiView transformer Assran et al. (2022), training remains computationally intensive. Using 2×A100 80GB GPUs, the total training time (excluding side information) is approximately 8 hours when training for 30 epochs.

### E.3 ABLATIONS

Table 14 (left) shows validation performance of the baseline (0% negatives) under different combinations of epochs, masking ratio (MR), and ME-MAX regularization strength ($\lambda$). Table 14 (right) compares performance under standard vs. stronger ME-MAX regularization, showing that MSN-SI benefits from increased regularization.

Table 14: Overview of baseline (0% negatives) hyperparameter search (left) and ME-MAX regularizer strength ($\lambda$) over different negative ratios (right). All numbers are from the validation set of PolypsSet.

(a) Hyperparameter Search

| MR(%) | Epochs | $\lambda$ | F1 |
|---|---|---|---|
| 50 | 30 | 1 | 77.1 |
| 75 | 10 | 1 | 75.5 |
| 75 | 30 | 1 | 77.4 |
| 75 | 30 | 3 | 78.3 |
| 75 | 50 | 1 | 73.8 |

(b) ME-MAX Regularization

| Method | $\lambda$ | 0 | 12.5 | 25 | 50 |
|---|---|---|---|---|---|
| MSN-N | 1 | 77.4 | 78.6 | 75.3 | 74.9 |
| | 3 | 78.3 | 77.3 | 75.6 | – |
| MSN-SI | 1 | – | 73.3 | 74.70 | – |
| | 3 | – | 75.8 | 78.2 | 77.9 |

