# OpenReview forum: "Self-Supervised Learning with Side Information"
_ICLR.cc/2026/Conference — Submitted to ICLR 2026_

### Official Review · Reviewer_5eJ7 · 2025-10-23

**Soundness:** 3
**Presentation:** 4
**Contribution:** 3
**Rating:** 4
**Confidence:** 5

**Summary:**

The paper tackles an important limitation of mainstream self-supervised learning: the Multi-View (MV) assumption treats all features shared across augmented views as useful, even when many are task-irrelevant nuisances. Therefore, it proposes a new Nuisance-Free Multi-View assumption that explicitly excludes nuisance variables. Specifically, it implements this framework using side information—auxiliary data that shares nuisance structure but lacks task-relevant information—and penalize representational overlap using a Jensen-Shannon divergence between main and side representations. However, the theoretical advance is incremental and the experimental breadth is insufficient to assert broad applicability.

**Strengths:**

1.	It cleanly formalises nuisance in an information-theoretic way and gives a generalisation bound that rewards minimal representations for the task subset, not all MV tasks.
2.	One extra term (JSD) with no additional encoders, or costly MI estimators.
3.	Writing is good and easy to follow.

**Weaknesses:**

1.	CIFAR+MNIST is useful, yet the nuisance (CIFAR background) is low-dimensional and visually very distinct from the signal (MNIST digit). It is unclear whether the method survives nuisances that are semantically richer or partially correlated with the label.
2.	The scale-up ability is not discussed, since CIFAR and MNIST only contain 10 classes.
3.	No comparison with other contemporary medical-SSL methods such as M2CRL, FocusMAE, or SepCLR run on identical data splits. Reported numbers mix different backbones and pre-training sets, making fair comparison difficult.
4.	Sensitivity analyses is missing.

**Questions:**

1.	What happens if the side set is not label-free?
2.	How does γ scale with nuisance dimensionality? How sensitive is γ to different nuisance strengths?
3.	What if the side set itself contains some task-relevant signal? This method assumes a single encoder that must represent both nuisance and signal to disentangle them. For very high nuisance complexity this could increase the necessary encoder capacity, contradicting the “minimal representation” narrative.

---

> ### Author Response · Authors · 2025-11-20
> **Semantically Rich Nuisance and Partial Correlations | Scale-up Ability**
>
> We thank the Reviewer for the careful review and helpful comments. We appreciate questions around how NF-MV behaves with richer nuisances, how it scales, and how to interpret the empirical comparisons. Below we respond point-by-point, and we will also update the main text to better highlight the relevant ablations and the intent of our evaluation.
>
> ### W1: On More Semantically Rich Nuisance and Partial Correlations.
> We agree that the CIFAR10+MNIST construction is a controlled setting, and we chose it precisely to isolate a foreground-background spurious correlation under experimental control. Importantly, the NF-MV principle is not tied to low-dimensional nuisances: it only assumes that some nuisance structure is shared across views and that side information provides an operational proxy for this structure. The colonoscopy setting directly complements the synthetic experiment in this regard. REAL-Colon contains high-dimensional, rich nuisances (mucosal texture, illumination changes, specularities, motion blur, and scope artefacts) acquired from the same procedures, patients, and devices as the main data. These are not trivially separable from polyp-positive frames, and their correlation with labels is weaker. The fact that NF-MV consistently improves over the same JEA baselines in this regime demonstrates that the framework survives beyond visually distinct, low-dimensional nuisance and remains effective under realistic nuisance complexity and partial correlations.
>
> ### W2: Scale-up Ability.
> The NF-MV is label-agnostic during pre-training. The JSD term separates main vs. side domains through their feature distributions, independent of the downstream label space or its cardinality. Consequently, neither the objective nor the theoretical guarantees depend on the number of classes, but rather on the information content of the learned representation and on whether nuisance structure is shared across views. In this sense, CIFAR+MNIST is not used to claim coverage of large label spaces, but to demonstrate the core MV failure mode and the corrective effect of NF-MV. The REAL-Colon results then show that the same principle transfers to a substantially more complex, large-scale medical domain.

---

> ### Author Response · Authors · 2025-11-20
> **Comparison with Medical SSL Methods | Sensitivity Analyses**
>
> ### W3: Comparison with Medical SSL Methods.
> We agree that comparisons to additional medical SSL baselines such as M2CRL, FocusMAE, or SepCLR on identical data splits would be informative. Both M2CRL and FocusMAE are video-based frameworks, and both are focusing on high-level colonoscopy tasks (detection, segmentation) rather than polyp histology and morphology classification. Since they additionally use other datasets for pretraining it is difficult to make a fair comparison. A comparison with SepCLR would be advantageous. However, in order to make fair comparison, substantial hyper-parameter tuning would be needed to adapt it to the colonoscopy domain. This was not feasible, as the dual encoder architecture makes it highly compute-intensive at this large scale. We will strive to implement these comparisons in future work in the "colonoscopy image analysis" space.
>
> Our current design focuses on two complementary aspects:
>
> - *Within-framework comparison.* We provide strong JEA baselines using exactly the same backbone and data as our method: for REAL-Colon, the MSN row in Table~3 is our baseline (ViT-S, same pre-training set, no side information), while MSN-N adds side data without NF-MV, and MSN-SI (ours) applies the JSD penalty on the identical setup. This isolates the contribution of NF-MV and side-information modelling, independent of architectural confounders.
> - *Comparison to prior medical SSL work.* We also compare to methods such as Hirsch et al. that use a larger private pre-training dataset and, in some cases, larger architectures. This shows that NF-MV with the public REAL-Colon can match models trained on much larger private corpora, highlighting the possible data-efficiency benefits.
>
> ### W4: Sensitivity Analyses.
> We agree that sensitivity analysis is important, and we actually include several such studies, which we will highlight more clearly in the main text:
>
> - *Side information ratio $R_{SI}$.* Tables 4, 7, and 8 systematically vary the fraction of side information in each batch across both natural-image and colonoscopy experiments. We find that performance is robust across a fairly wide range, with $R_{SI} \approx 25\%$ performs well across both domains.
> - *JSD weight $\gamma$.* Table 2 (and the corresponding ablations in the appendix) examines different values of $\gamma$. Larger values strengthen nuisance separation but can lead to instability if the JSD term dominates the base SSL loss; in practice we find a broad region where performance is stable and improved over the baseline.
> - *Side information quality.* Table 9 investigates controlled contamination of the side dataset, injecting varying levels of label-relevant signal. Performance degrades gracefully as contamination increases and remains above the baseline for moderate impurity levels, indicating that NF-MV is not brittle.

---

> ### Author Response · Authors · 2025-11-20
> **Side-set not Label-Free | JSD strength and Nuisance Dimensionality | Capacity and Minimal Representations**
>
> ### Q1: Side-set not Label-Free.
>
> The Reviewer raises an important point: in many real-world scenarios, the side set cannot be guaranteed to be perfectly label-free, and we agree that a perfectly label-free side set can be an idealised assumption. NF-MV only requires side information to be approximately task-irrelevant while sharing nuisance structure with the main data. We test this failure mode in Table~9 by injecting controlled amounts of task-relevant signal into the side set. The results show that NF-MV yields gains when the side set is clean or mildly contaminated; as contamination increases, the benefit decreases rather than collapsing. Intuitively, if side data increasingly overlaps with main-task content, the JSD term carries less useful nuisance contrast and the objective introduces more noise. Thus, moderate impurity does not seem to invalidate NF-MV, but reduces its advantage.
>
> ### Q2: JSD strength and Nuisance Dimensionality.
> The parameter $\gamma$ is a relative weighting that trades off the base SSL loss against the JSD separation term (identifying nuisance via main--side contrast). Although we do not believe this strength is directly related to the nuisance dimensionality, it can be seen as the importance of separating main and side representations, relative to the importance of aligning views. There is however reason to believe that the nuisance complexity relates to the capacity of the network and quantity of side information, which we discuss below. In case this does not answer the question, we would kindly appreciate clarification.
>
> ### Q3: Capacity and Minimal Representations.
> We thank the Reviewer for raising this important point. NF-MV does require the encoder to capture nuisance structure enough to identify it, otherwise it cannot be discarded, so in regimes where nuisance is extremely complex, additional encoder capacity may be beneficial. This, however, is not in contradiction with a minimal representation narrative. The key distinction is that minimality refers to the information content of the learned representation $Z$, not to the raw capacity of the encoder network. These notions are different: a model may have high capacity yet still learn a focused, compressed representation. A larger encoder provides more flexibility to extract the appropriate information; it does not inherently force $Z$ to retain more information about the input.
>
> Relatedly, the side-information ratio $R_{SI}$ determines what fraction of each batch comes from the side set, and thus how much gradient budget is allocated to learning the side data relative to the base SSL objective on main data. It is important to keep in mind that the nuisance is assumed to exist in the main data as well. Our ablations varying $R_{SI}$ (Tables~4, 7, 8) show that a modest value, typically around $R_{SI} \approx 25$%, performs best: it provides sufficient nuisance signal to guide learning and allow separation, without letting side-modelling dominate training.
>
> This ties back to the nuisance complexity the Reviewer mentioned. If we assume the nuisance complexity is very high and difficult to learn, we would like to increase $R_{SI}$ to a point where the nuisance is sufficiently well represented, and this takes away from learning the main data. Importantly, high nuisance complexity is also precisely the regime where standard MultiView SSL fail. When nuisance dominates what is shared across views, a regular JEA will allocate capacity to those shared nuisances anyway, drowning out task-relevant structure. NF-MV does not introduce a new dependency on nuisance complexity so much as provide a controlled mechanism to identify and separate the nuisance that the model would otherwise encode by default.

---

> ### Comment · Reviewer_5eJ7 · 2025-11-24
> **Further suggestion**
>
> Thanks for the clarification. And there are some further suggestions：
> 1.	Increase background dimension (e.g., 32×32 → 128×128)
> 2.	Backbone of ViT-B and ViT-L (instead of ViT-S) should be included to show the scale-up ability of NF-MV
> 3.	Provide ε-bound: how much I(Y;X_side) is tolerated before gain ≤ 0.

---

> ### Author Response · Authors · 2025-12-03
>
> We thank the Reviewer for their reply. We trust the previous concerns have been addressed. Below, we address their additional suggestions.
>
> ### (1, 2) Higher-dimensional backgrounds in controlled experiments and larger backbones for the colonoscopy data encoder.
>
> Unfortunately, we could not run also these additional, computationally-onerous  experiments within the limited discussion period. However, we want to point out that NF-MV does not depend on pixel dimensionality but on shared nuisance structure and the availability of side information. In fact, the colonoscopy experiments already operate in a high-dimensional, semantically rich nuisance regime and show consistent gains. Moreover, the paper now includes experiments of integrating our JSD term into five different SSL approaches, suggesting that our approach is architecture-agnostic. We will include this discussion in the paper.
>
> ### (3) Tolerance to label contamination in the side set.
> A formal $\varepsilon$-bound of the form $I(Y;X_{\text{side}}) \leq \varepsilon$ is theoretically appealing, but difficult in general, as the bound would depend on the instantiation of the NF-MV assumption. Although not a bound, we below show how the proposed JSD instantiation of NF-MV qualitatively behaves under side information contamination.
>
> We begin by considering the behaviour of the JSD penalty in a limiting case of contamination. The JSD penalty promotes separation between $p_\theta(z \mid X_{\text{main}})$ and $p_\theta(z \mid X_{\text{side}})$; as contamination grows, these two feature distributions move closer. In the limiting case where $X_{\text{main}}$ and $X_{\text{side}}$ are drawn from the same input distribution, we have $p_\theta(z \mid X_{\text{main}}) = p_\theta(z \mid X_{\text{side}})$ for all $\theta$, and hence
> $$\mathrm{JSD}\big(p_\theta(z \mid X_{\text{main}})\,\|\,p_\theta(z \mid X_{\text{side}})\big) \equiv 0$$
> so its population gradient vanishes, $\nabla_\theta \mathrm{JSD} = 0$. In practice we optimise a minibatch estimator, whose expected gradient is then zero and whose residual contribution is due to finite-sample noise. This explains why the operational method (JSD) is robust to contamination of side information: it introduces additional noise into the gradients, but does not collapse the objective.
>
> The analysis can be extended to the partial contamination case where $P_\theta := p_\theta(z \mid X_{\mathrm{main}})$ and $Q_\theta := p_\theta(z \mid X_{\mathrm{side}})$ become close. In this regime the log-density ratios $\log(P_\theta/M_\theta)$ and $\log(Q_\theta/M_\theta)$ shrink towards zero, and since JSD is an $f$-divergence, its population gradient vanishes quadratically in the distributional difference. A minibatch estimator, however, replaces the expectations by averages over a fixed batch size $B$, so its variability is determined by finite-sample fluctuations of these log-ratio terms. These fluctuations decrease only through the usual $1/\sqrt{B}$ scaling of sample means and therefore do not vanish at the same rate as the population gradient when $P_\theta \to Q_\theta$. As a result, the *noise-to-signal ratio* of the JSD gradient increases as contamination grows; in the limit $P_\theta = Q_\theta$, the signal disappears while the estimator reduces to pure sampling noise. This explains why under contamination the JSD term becomes increasingly noisy, possibly hurting training dynamics.
>
> This qualitative picture is consistent with our empirical study (Table 9), where performance degrades as side information becomes more contaminated, rather than collapsing learning. We will include this derivation in the appendix of the paper.

---

### Official Review · Reviewer_WZCc · 2025-10-28

**Soundness:** 4
**Presentation:** 3
**Contribution:** 3
**Rating:** 8
**Confidence:** 4

**Summary:**

- This paper critiques the standard multi view asusmpting in self-supervisied learning for being too permissive, allowing task-irrelevant nuisance factors, such as dominant background textures in colonoscopy, to persist across views and become entangled with task-relevant signals.
- The authors introduce a Nuisance-Free MultiView (NF-MV) assumption, which reframes the objective of self-supervised learning (SSL) as learning representations sufficient for the task while being explicitly invariant to shared nuisance structure.
- The nuisance penalty is implemented practically by approximating an idealized mutual information objective using the Jensen-Shannon divergence (JSD) between the representations of the main and side data distributions.
- The authors provide a simple, modular, and architecture-agnostic extension to standard Joint Embedding Architectures (JEAs) like Barlow Twins, CorInfoMax, and Masked Siamese Networks (MSN), introducing negligible computational overhead.
- Experiments on controlled Cifar10+MNIST datasets, engineered with spurious correlations, demonstrated that the side information (SI) variants significantly improved performance on generalization tasks compared to standard and naive baselines.

**Strengths:**

- Strong conceptual contribution: The formalization of the Nuisance-Free MultiView (NF-MV) assumption is a solid conceptual contribution. It provides a new and useful information-theoretic perspective on what SSL should (and should not) be learning, backed by a theoretical argument for improved generalization. The NF-MV assumption provides a principled extension of the standard MultiView framework that explicitly models what representations should not preserve. Theorem 1 formalizes the generalization benefit of learning minimal sufficient representations for the target task rather than all MultiView-induced tasks, adapting the Xu–Raginsky bound to show strictly tighter generalization bounds when `I(Z'; X) < I(Z; X)`.
- Simple and general: The proposed method is elegant in its simplicity. It is not a complex new architecture but a modular penalty term that can be added to existing joint embedding architectures. The paper contrasts this favorably against contrastive analysis methods like SepCLR, which require multiple separate encoders (one for salient features, one for common features), multiple feature spaces, and substantially higher computational and memory costs.
- Strong emperical results: There are two sets of experiments in the paper, on a hybrid synthetic Cifar+MNIST, and on real-world colonoscopy data.
  - The Cifar+MNIST experiments provide clear evidence that the method helps models overcome spurious correlations.
  - On the colonoscopy experiments with PolypsSet, this methods achieves 80.3% F1 on histology classification while matching models trained on an order of magnitude more private data. Clearly demonstrates the value of leveraging typically discarded background frames as side information.

**Weaknesses:**

- Limited guidance on side information selection: The Nuisance-Free MultiView (NF-MV) assumption strictly requires that the nuisance variable n is independent of the label y, formalized as `I(y;n)=0`. This assumption is restrictive in real-world scenarios, and the paper lists this precise point as a core limitation. The authors provide limited guidance on systematically identifying appropriate side information sources beyond the colonoscopy example. How would practitioners determine whether available auxiliary data satisfies this assumption? How much contamination is tolerable before performance degrades below baseline?
- Coarse mutual information proxy: One weakness is in the practical implementation of the JSD penalty. While the theoretical objective aims to maximize the complex mutual information $\mathbf{I}(\mathbf{z}; B_\alpha)$, the practical implementation computes the JSD between the average softmax outputs of the main and side batches. The authors note this, and it clearly works, but it feels disconnected from the information-theoretic motivation and may not be the most powerful way to enforce separation. The authors provides no empirical validation of approximation quality, such as, how well does $\text{JSD}_\alpha(\mathbf{\bar{z}}_X || \mathbf{\bar{z}}_S)$ correlate with $\mathbf{I}(\mathbf{z}; B)$ estimated via methods like CLUB or InfoNCE? Given that the cited MI estimators (lines 257-262) are dismissed for "high variance and bias" without demonstration, it's unclear whether the simplicity+tractability tradeoff is optimal or whether intermediate approaches might be more effective.
- Minor point on hyperparameter guidance: While the `γ` values differ across methods (reflecting their different loss scales), the paper could provide a heuristic for initial `γ` selection—perhaps as a ratio to the primary SSL loss magnitude. However, the hyperparameter search required is modest and shows predictable trends.

**Questions:**

- Can you provide more systematic guidance or a principled procedure for identifying valid side information in new domains? What diagnostic tests or metrics could practitioners use to verify the `I(y;n)=0` assumption? (really, just restating my weakness point).
- Could you confirm that the "MSN" row (76.1, avg F1) is your own controlled baseline, using the same ViT-S architecture and pre-trained on the same REAL-Colon dataset as your method, just without the side information? And that the "MSN-SI (ours)" row (80.3, avg F1) is the identical setup plus your JSD penalty? The table formatting doesn't clearly distinguish their baseline from prior work.

---

> ### Author Response · Authors · 2025-11-20
> **Side Information Selection | JSD as Proxy**
>
> We thank the Reviewer for the positive evaluation and for engaging with both the theoretical framing and the practicality of the method. We address each raised weakness/question below.
>
> ### W1/Q1: Side Information Selection.
> We agree that the NF–MV assumption $I(y;n)=0$ is strong, and it is an idealised modelling assumption. In practice we need the side set to be approximately task-irrelevant. In Table 9 we investigate how corrupting the side information affects the performance, seeing that performance degrades gracefully when introducing task-relevant information into the side information. Regarding how to select side information, our view is that this is domain-specific: while this can be seen as a limitation, we consider it as a potential strength too, as it allows to make use of domain knowledge when training self-supervised models (JEAs), something that is otherwise limited to handcrafting augmentation policies that are suitable for learning representations useful for the downstream tasks. In practice, the choice of side information can be quite natural. In colonoscopy, polyp-negative frames share imaging nuisances (mucosal texture, illumination changes, motion, specularities) with the main data but lack polyp semantics, making them a good operational proxy for $n$. In other domains, similar “background-heavy” or “non-event” pools often exist (e.g., off-task frames in videos, non-object regions in detection datasets, or sensor readings outside event windows). When such a clear split is not available, an interesting approach is to leverage anomaly- or event-detection models (possibly trained on a small labelled subset or even unsupervised) to partition a large pool into “candidate main” (likely event/anomaly) and “candidate side” (likely background) subsets. Our CIFAR+MNIST results and the contamination experiment (Table 9) indicate that this partition does not need to be perfect for NF–MV to provide gains: approximate nuisance-richness is sufficient. Yet, this is an area future research: error-propagation and bias from the anomaly detection algorithm needs to be analysed and handled carefully.
>
> ### W2: JSD as Proxy
> We agree with the Reviewer in that the connection between the ideal mutual information objective and our JSD penalty deserves clearer exposition. Our goal is to maximize $I(Z;B)$, where $z = f(x)$ and $B$ indicates the sample origin. We show that this can be written as a (weighted) Jensen-Shannon divergence between the conditional feature distributions (eq. 2). These distributions are difficult to estimate in high-dimensions, so we introduce a proxy by passing $z$ through a softmax head and treat the output as a distributions over discrete pseudo-labels. If we define the discrete variable $Y$ via $\operatorname{Pr}(Y=y | \sigma(z))$, the batch-averaged softmax vectors are then the Monte-Carlo estimates of the domain-conditional label distribution $\operatorname{Pr}(Y=y | B=0)$ and $\operatorname{Pr}(Y=y | B=1)$. It follows that $I(Y;B) = \operatorname{JSD}(\mathbb{E}[\sigma(z)]|B=0) || \mathbb{E}[\sigma(z)]|B=1))$. By the data processing inequality, $I(Y;B) \leq I(Z;B)$, so the JSD term can be seen as a tractable lower bound on the mutual information, which we maximize. There is however no guarantee that this is a tight bound, due to the coarseness of the approximation, which is a limitation with the current approach. For the revised version of the paper, will add this motivation for the use of JSD and batch-averaged predictions.

---

> ### Author Response · Authors · 2025-11-20
> **Hyperparameter Guidance | MSN Table**
>
> ### W3: Hyperparameter Guidance.
> We agree with the Reviewer that further discussion about hyperparameters would strengthen the intuition and reproducibility of the method, and will dedicate more room in the main paper for this. We investigate side information ratio $R_{SI}$ for both the colonoscopy and natural image experiments (Tables 4, 7, 8), and found that modest amounts $R_{SI} = 25$% often performs well across both domains. The strength of the JSD term depends on the scale of the overall loss: generally a high $\gamma$ performs well (Table 2), but it can make the loss unstable if coupled with also a high $R_{SI}$ (Tables 7 and 8).
>
> ### Q2: MSN Table.
> Yes, the Reviewer's interpretation is correct. The MSN row (76.1 avg F1) is our own controlled baseline: a Masked Siamese Network with a ViT-S backbone, pre-trained on REAL-Colon using only polyp-positive data, without using side information or the JSD penalty. The optimisation hyperparameters, architecture, and augmentations are those detailed in Appendix E. Substantial efforts and hyperparameter-tuning were dedicated to creating this baseline. The MSN-SI (ours) row (80.3 avg F1) uses the same ViT-S backbone, pre-training dataset (REAL-Colon), and base optimization parameters, but additionally incorporates side information and our JSD-based NF–MV objective.

---

### Official Review · Reviewer_wWiT · 2025-10-31

**Soundness:** 3
**Presentation:** 3
**Contribution:** 2
**Rating:** 4
**Confidence:** 4

**Summary:**

This paper presents the Nuisance-Free MultiView (NF-MV) framework for self-supervised learning (SSL), which argues that standard MultiView-based SSL methods (e.g., SimCLR, Barlow Twins, MSN) are too permissive because they retain shared but task-irrelevant features. To address this, the authors propose incorporating side information—data that shares nuisance structure but lacks task-relevant signals—and penalizing representational overlap with a Jensen–Shannon divergence (JSD) term. Experiments on synthetic CIFAR+MNIST tasks and colonoscopy datasets show improved generalization.

**Strengths:**

- Identifies a real and well-documented weakness in existing self-supervised learning (SSL): the MultiView assumption often retains shared nuisance features that hurt downstream generalization.
- Provides a formal, information-theoretic reformulation through the Nuisance-Free MultiView (NF-MV) assumption, which is explicitly stated and mathematically well-defined.
- Works seamlessly with multiple Joint-Embedding Architectures (JEAs): Barlow Twins, CorInfoMax, Masked Siamese Networks, etc.
- Computationally lightweight: the JSD is computed from existing model outputs (softmaigible overhead.

**Weaknesses:**

- The paper’s central claim, that SSL can be improved by discouraging shared nuisance structure, is not new. The idea of separating task-relevant vs nuisance components dates back to Domain Separation Networks, Contrastive Analysis, and information bottleneck extensions. The NF-MV assumption is a relabeling of those ideas with minor formal changes; it adds no new theoretical insight beyond replacing the nuisance variable n with a side dataset S. The theoretical results (Theorem 1) are direct corollaries of Xu & Raginsky (2017) and do not provide novel analysis or guarantees specific to SSL.
- Method is heuristic and poorly justified theoretically. The proposed JSD penalty is applied to batch-averaged prototype softmax distributions rather than instance-level representations. This design is ad hoc and undermines the information-theoretic motivation, as it does not properly estimate mutual information. The approach maximizes separation (JSD) between main and side data rather than explicitly enforcing invariance to nuisances. The desired disentanglement of task and nuisance subspaces is therefore not guaranteed, and may in fact promote encoding of domain cues rather than removal of them. The method’s behavior depends critically on the quality and purity of the side data, which is not verifiable or robustly handled.
- Weak empirical validation. The synthetic CIFAR10+MNIST setup is a toy problem where the nuisance (background) is obvious and separable, offering limited insight into real-world complexity.
In the colonoscopy experiments, the side data (negative frames) and target data (positive frames) are trivially distinct in pixel distribution. Thus, improvements may stem from easier domain separation rather than true nuisance invariance.
Baselines are unevenly compared: SepCLR is trained for half as many epochs (500 vs 1000), and private datasets used by prior works differ in size and quality.
No other domains (e.g., natural videos, non-medical images) are tested, so generalization is unproven.
- Limited impact and scope. The method’s benefit depends on having large quantities of “nuisance-only” data (e.g., 87% negative frames in REAL-Colon). This is uncommon outside of specific medical-imaging domains.
Without clear theoretical innovation or strong empirical breadth, the contribution is too incremental for a top-tier venue.

**Questions:**

In open-domain SSL (e.g., ImageNet or LAION pretraining), how can “side information” be realistically defined and obtained? Specifically, what data could serve as a nuisance-only side set when task relevance is unknown—how would you verify its purity, scale it to diverse domains, and avoid suppressing features that later prove useful for downstream tasks?

---

> ### Author Response · Authors · 2025-11-20
> **Theoretical Insights.**
>
> We sincerely thank the Reviewer for the careful assessment. We appreciate the opportunity to clarify our intended setting and the theoretical role of NF-MV. The aim of this work is not to claim that side information is always available or that NF-MV should be used universally. Rather, we present a principled formulation and practical method for settings where side information is present or can be reasonably identified, and where one care about learning focused representations aligned with the intended use of those representations. We argue NF-MV offers a concrete way to leverage such side data to avoid spurious, nuisance-driven dependencies. This is in contrast with open-domain SSL or foundation models, where the goal is to learn representations that are wide and general.
>
> ### W1: Theoretical Insights.
> We thank the Reviewer for this observation and agree that the high-level idea of separating task-relevant from nuisance structure is not new. Our contribution is not in introducing this concept, but in its characterization and formalization for the wide family joint-embedding architectures (JEAs) in self-supervised learning.
>
> First, the NF-MV assumption is not merely a relabelling of previous nuisance formulations. It provides a refinement of the classical MultiView assumption by decomposing the MultiView-induced task family $\mathcal{T}$ into a stricter subset $\mathcal{T}_{\mathrm{nf}}(n)$ consisting of tasks that are simultaneously MultiView-compatible and independent of nuisance structure. NF-MV allows us to specify which subset of tasks should be preserved, which then allows us to apply the generalization bound in new ways. Second, while Theorem 1 indeed builds upon Xu & Raginsky, its novelty lies in applying their generalization bound to representations that are minimal sufficient for the full MultiView task family, versus the stricter NF-MV family. To our knowledge, this connection between MultiView structure, nuisance modelling, and representation minimality in JEAs has not been previously developed. We therefore do not claim to introduce a new generalization bound, but rather to provide a new interpretation and application of existing information-theoretic tools tailored to joint embedding methods. This was possible through the establishment of NF-MV, which we further believe motivates the new perspective and theoretical contribution.
> For comparison, the work by Wang et al. (CVPR 2022), which explored the opposite but complementary case where task-relevant information is not fully shared across views, was likewise a focused refinement of the standard MultiView formulation, yet introduced an important shift in perspective on MultiView learning. Similarly, our NF-MV assumption highlights a limitation in the overly permissive nature of standard MultiView SSL. We believe this contribution is substantive and timely.

---

> ### Author Response · Authors · 2025-11-20
> **JSD as Heuristic**
>
> ### W2: JSD as Heuristic.
>
> We appreciate these concerns and will clarify the theoretical motivation, and limitations, of our operational approach.
>
> (i) We agree with the Reviewer in that the connection between the ideal mutual information objective and our JSD penalty deserves clearer exposition, especially the use of batch-averaged predictions. Our goal is to maximize $I(Z;B)$, where $z = f(x)$ and $B$ indicates the sample origin. We show that this can be written as a (weighted) Jensen-Shannon divergence between the conditional feature distributions (eq. 2). These distributions are difficult to estimate in high-dimensions, so we introduce a proxy by passing $z$ through a softmax head and treat the outputs as distributions over discrete pseudo-labels. If we define the discrete variable $Y$ via $\operatorname{Pr}(Y=y | \sigma(z))$, the batch-averaged softmax vectors are the Monte-Carlo estimates of the domain-conditional label distribution $\operatorname{Pr}(Y=y | B=0)$ and $\operatorname{Pr}(Y=y | B=1)$. It follows that $I(Y;B) = \operatorname{JSD}( \mathbb{E}[\sigma(z)|B=0] || \mathbb{E}[\sigma(z)|B=1] )$. By the data processing inequality, $I(Y;B) \leq I(Z;B)$, so the JSD term can be seen as a tractable lower bound on the mutual information, which me maximize. There is however no guarantee that this is a tight bound, due to the coarseness of the approximation, which is a limitation with the current approach.
>
> (ii) The JSD penalty is a nuisance identification and separation mechanism.
> Our goal is not to encode nuisance structure for its own sake. Rather, identifying nuisance structure is necessary step for determining what should be disregarded from main representations. The side dataset $\mathcal{S}$ provides a concrete means of isolating what varies across domains but is irrelevant to the target task family. The JSD penalty encourages the encoder to respond differently to $\mathcal{X}$ (the main dataset) and $\mathcal{S}$, thereby identifying and focusing on what separates the domains. The main SSL objective and downstream task subsequently emphasize task-relevant structure. This is aligned with contrastive analysis and domain-adversarial approaches.
>
> (iii) Dependence on side-data quality.
> We agree that understanding how performance scales with both the quality of side information is an important aspect, and central to the practical deployment of NF-MV. Our goal in this paper is to establish the core principle that side information can serve as an operational approximation to nuisance structure to improve self-supervised learning; nevertheless, we have taken initial steps toward characterising these effects. In the controlled CIFAR experiments, Table 9 investigates the impact of imperfect side information by injecting controlled levels of label noise into the side dataset. This demonstrates that the method does not require a perfectly curated nuisance distribution, and that some degree of overlap or contamination does not invalidate the NF-MV objective. However, we emphasize that this remains a controlled setting designed to isolate specific failure modes. In real-world applications, the notion of quality is inherently more complex. That said, we view a more thorough investigation of side information quality and scaling behaviour-as an important direction for future work. The present results provide encouraging evidence that NF-MV is not brittle with respect to moderate imperfections in side information.

---

> ### Author Response · Authors · 2025-11-20
> **Empirical Validation**
>
> ### W3: Empirical Validation.
> The creation of the synthetic dataset is done to perform controlled experiments in an isolated setting, allowing us to assess the impact of our approach before moving to real-world scenarios where multiple confounding factors are at play. We also believe it to highlight an important limitation of MultiView learning. We do not believe the controlled experiment is a simple problem, where nuisance and target data are trivially distinct. Although the overlaid MNIST digit comes from another pixel distribution, it affects a small amount of pixels, and the picture is dominated by the more complex background. In fact, the experiments show that standard joint embedding methods collapse in this setting. Additionally, if separating between main and side samples was trivial, this separation would not encourage the model to learn discriminative features between the MNIST digits better, which the experiments show that our proposed method does. The colonoscopy experiments then show that these improvements carry over to a substantially more complex domain, where main and side data are definitely neither easily separable nor come from trivially distinct pixel distributions: polyp detection is difficult task and there are many polyp-like structures that appear during colonoscopy. Additionally, both main and side frames are acquired from the same patients, endoscopes and centres, and share most low-level statistics (mucosal texture, illumination). This is also demonstrated by the fact that all models for polyp detection typically struggle with false polyps. Together this showcases that neither the controlled or colonoscopy experiments are easy, or that the main and side information can be trivially separated.
>
> SepCLR’s 500 epochs require similar compute to 1000 epochs of Barlow Twins/CorInfoMax due to its dual encoders, and we further saw it reaching a plateau. We also compared against naive inclusion of side data, which failed to improve performance. For the colonoscopy experiments, we provide strong JEA baselines using exactly the same backbone and data as our method: for REAL-Colon, the MSN row in Table 3 is our baseline (ViT-S, same pre-training set, no side information), while MSN-N adds side data without NF-MV, and MSN-SI (ours) applies the JSD penalty on the identical setup. This isolates the contribution of NF-MV and side-information modelling, independent of other methodological confounders and datasets. When comparing to prior work we compare to methods such as Hirsch et al. that use larger private pre-training datasets and, in some cases, larger architectures. We are explicit that these are not perfect comparisons, yet they show that NF-MV with the public REAL-Colon can match the performance of models trained on a larger private corpora.

---

> ### Author Response · Authors · 2025-11-20
> **Nuisance Availability, Impact, and Scope | Open-domain SSL**
>
> ### W4: Nuisance Availability, Impact, and Scope.
> We respectfully disagree with the premise that real-world settings rarely contain side information. In practice, the opposite is typical: most naturally collected visual data are not tightly object-centric, whereas the datasets commonly used in deep learning are heavily curated toward the downstream label of interest. Medical imaging provides many concrete examples. Across dermatology, pathology/tissue microscopy, radiology, and endoscopy, large fractions of data contain no visible finding (e.g., normal frames/regions) even though they are collected as part of the same workflow. In screening and routine acquisition, “no-finding” images are abundant and are often easier to detect than the fine-grained diagnosis that follows; these images therefore constitute natural side information for the downstream classification task. This structure also appears well beyond healthcare. In industrial inspection, identifying whether any defect is present is typically simpler than classifying defect type or severity, and production lines naturally generate large volumes of normal surfaces. Similarly, in retail and surveillance, detecting “person/customer present vs. empty scene” is easier than recognizing specific activities or attributes, while empty or irrelevant frames dominate continuous capture.
>
> These examples illustrate that nuisance/side images are not an edge case but a common byproduct of real data collection, and methods that can exploit such side information can have a broad impact.
>
> ### Q1: Open-domain SSL.
> We would like to stress that our approach is not developed for open-domain SSL. In open-domain SSL, the goal is to learn wide representations that are useful for as many tasks as possible, and this is indeed useful when there is little or no information about which downstream tasks will later be of interest. This is in a way opposite to our approach, as we are targeting discriminative representation learning, i.e. learning representations that are useful for the "relevant tasks". A consequence of this is that one should know what these relevant tasks are. Actually, we believe this to be one of the more important contributions of our work: the NF-MV perspective provides a way to naturally include task and domain knowledge into MultiView learning, something that is otherwise done through handcrafting augmentation policies.

---

### Official Review · Reviewer_NwSZ · 2025-11-01

**Soundness:** 4
**Presentation:** 4
**Contribution:** 4
**Rating:** 6
**Confidence:** 4

**Summary:**

This paper proposes an extension to the standard MultiView assumption in self-supervised learning (SSL) by introducing the Nuisance-Free MultiView (NF-MV) assumption, which aims to make representations invariant to task-irrelevant (nuisance) factors while preserving task-relevant information. The authors motivate this with information-theoretic analysis, drawing from the Information Bottleneck principle, and implement it practically by leveraging "side information"—auxiliary data that shares nuisance structures but lacks task signals. They approximate a mutual information penalty using a Jensen-Shannon divergence (JSD) between representations from main and side data, making it compatible with joint embedding architectures (JEAs). Experiments on a synthetic Cifar10+MNIST dataset with spurious correlations and real-world colonoscopy data (using REAL-Colon) show improvements in generalization for methods like Barlow Twins, CorInfoMax, and Masked Siamese Networks.

**Strengths:**

1. The information-theoretic framing is solid and builds nicely on prior work like the MultiView assumption (Sridharan & Kakade, 2008) and extensions of the Information Bottleneck (Chechik & Tishby, 2002)
2.The JSD-based penalty is lightweight, adding negligible compute overhead, and plugs into existing JEAs without needing multiple encoders (unlike contrastive analysis methods like SepCLR). This modularity is appealing for real-world adoption.
3. The writing is clear, with good figures (e.g., Fig. 1 illustrating info overlap). It opens a perspective on explicitly modeling what not to learn in SSL, which could apply beyond medical imaging to domains with persistent nuisances.

**Weaknesses:**

1. The NF-MV assumption relies on well-defined side information that's nuisance-rich but task-irrelevant. In the colonoscopy case, polyp-negative frames work well, but the paper doesn't deeply explore how to identify or generate such data in less obvious domains. What if side info inadvertently includes subtle task signals?
2. While the synthetic and real experiments are solid, they're somewhat narrow. Only a few JEAs are tested (Barlow Twins, CorInfoMax, MSN), and no comparisons to other debiasing techniques like invariant risk minimization or recent SSL variants (e.g., DINOv2 or MAE adaptations). The colonoscopy results are promising, but use a single dataset for pre-training; cross-dataset validation (e.g., on KVASIR or EndoScene) would strengthen claims.
3. The JSD weighting (α, γ) and side info ratio (e.g., 5-40%) seem tuned per experiment, but ablation details are mostly in appendices. More discussion on robustness to these choices would help reproducibility.
4.How does performance scale with side info quality/quantity?

**Questions:**

See weak nesss
If my concern is resolved, I am willing to raise it to 8 points; this is a very novel work.

---

> ### Author Response · Authors · 2025-11-20
> **Nuisance Perspective & Side Information Availability**
>
> We thank the Reviewer for the thoughtful assessment, and for recognizing the novelty and potential impact of the NF-MV framework. We appreciate the constructive comments and questions raised. Below we address each point in detail.
>
> ### W1: Nuisance Perspective & Side Information Availability.
>
> What should be treated as main and side information is inherently domain-specific, so it is difficult to provide a one-size-fits-all answer to this very relevant question. However, the NF‑MV assumption is not tied to any specific type of nuisance. If, based on domain knowledge, one believes that most of the sample information is task-relevant, then defining nuisance factors may not be meaningful, reducing the method to standard MV. Conversely, when domain expertise can identify features that are task-irrelevant but correlated with labels, incorporating this knowledge into the NF‑MV framework provides a safeguard against learning spurious dependencies that harm robustness and generalization.
>
> In practice, most naturally collected visual data are not object-centric, whereas the datasets commonly used in deep learning are curated toward the downstream labels of interest. Medical imaging provides many concrete examples. Across dermatology, pathology/tissue microscopy, radiology, and endoscopy, large fractions of data contain no visible finding (showing only normal frames/regions) even though they are collected as part of the same workflow. In screening and routine acquisition, “no-finding” images are abundant and are often easier to detect than the fine-grained diagnosis that follows; these images therefore constitute natural side information for the downstream classification task. In industrial inspection, identifying whether any defect is present is typically simpler than classifying defect type or severity, and production lines naturally generate large volumes of normal surfaces. Similarly, in retail and surveillance, detecting “person/customer present vs. empty scene” is easier than recognizing specific activities or attributes, while empty or irrelevant frames dominate continuous capture. These examples illustrate that nuisance/side images are not an edge case but a common byproduct of real data collection, and methods that can exploit such side information can have a broad impact.
>
> If the available data does not readily include a split between "main" and "side" samples through high-level labels, there are interesting avenues to explore. In many domains, it is relatively easy to train anomaly detectors, either using a small set of labels or completely unsupervised. Under the mild assumption that anomaly detection is a simpler task than classifying the type of anomaly, this detector can be used to divide the larger unlabelled dataset in "main" and "side" information, showing anomalies and background, respectively. The results from Table 9 (where side information is corrupted with task-relevant signal) are promising in this regard: the anomaly detection algorithm does not need to be perfect for our framework to be advantageous. Yet, this is an area future research: error-propagation and bias from the anomaly detection algorithm needs to be analysed and handled carefully.

---

> ### Author Response · Authors · 2025-11-20
> **Empirical Scope**
>
> ### W2: Empirical Scope.
> We agree that the empirical evaluation focuses on a subset of joint-embedding methods, and we appreciate the Reviewer’s suggestion to broaden the comparison set. Our intention, however, is to isolate the contribution of side information within the NF-MV framework.  The focus of our paper is to show that the MultiView perspective, as used in the Joint Embedding Architectures of SSL, can be too permissive. To exhibit this, we believe it most natural to benchmark JEA models, and see how performance changes when implementing our new NF-MV perspective. The methods we have chosen to showcase this are deliberately selected from different sub-areas of joint embedding methods: Barlow Twins is a redundancy reduction method (similar to, for example VICReg) where feature dimensions are encouraged to be uncorrelated, CorInfoMax is an information maximization method, while MSN learns in a teacher-student framework (similar to BYOL, DINO). Together, they cover a wide range of modern JEA frameworks. Regarding additional cross-validation for the colonoscopy task, this would be desirable. However, public datasets that are large enough for SSL pre-training is not common. For the downstream performance, we use PolypsSet and the SUN database - two high-quality datasets for histology and morphology classification, respectively. Although there are other possible datasets for downstream validation, they are often heavily imbalanced or do not have the fine-grained classification labels needed.
>
> Masked image modelling (MIM) methods, such as MAE, operate under fundamentally different inductive biases. They rely on masking strategies to drive learning, instead of augmentations, and thus do not operate under the MultiView assumption which this work targets. Incorporating MIM architectures is an interesting direction, but orthogonal to the question addressed by NF-MV: how to exploit side information to handle nuisance structure in that persists across views in the MultiView learning employed by joint embedding methods.
>
> Standard IRM is primarily a supervised method that assumes labelled data and predefined environments, so it is not directly applicable to our self-supervised, side-information setting. An SSL adaptation such as IP-IRM is more relevant, since it applies an IRM-style invariance penalty to contrastive learning using discovered partitions as environments. However, IP-IRM’s partitions are not guided by side information or downstream tasks. As a result, when IP-IRM disentangles factors, it provides no guarantee that the remaining representation will privilege the task-salient (digit) feature over a still-correlated spurious one. In contrast, SepCLR and our approach use side/background data to define nuisance and suppress it from the representation of interest (e.g., SepCLR maps background to an informationless code in the salient space, our approach maximizes a divergence), so a downstream classifier is steered toward the digit feature even when the correlations persist. Because of this, methods such as IP-IRM would not make for a fair comparison.

---

> ### Author Response · Authors · 2025-11-20
> **Hyperparameter Discussion & Side Information Quality and Quantity**
>
> ### W3: Hyperparameter Discussion.
> We agree with the Reviewer that further discussion about hyperparameters would strengthen the intuition and reproducibility of the method, and will dedicate more room in the main paper for this. We investigate side information ratio $R_{SI}$ for both the colonoscopy and natural image experiments (Tables 4, 7, 8), and found that modest amounts $R_{SI} = 25$% often performs well across both domains. The parameter $\alpha$ is actually not tuned, but is a direct consequence of $R_{SI}$, as it is defined to be the probability that a sample comes from the main data. The strength of the JSD term depends on the scale of the overall loss: generally a high $\gamma$ performs well (Table 2), but it can make the loss unstable if coupled with also a high $R_{SI}$ (Tables 7 and 8).
>
> ### W4: Side Information Quality and Quantity.
> We agree that understanding how performance scales with both the quality and quantity of side information is an important question, and central to the practical deployment of NF-MV. Our goal in this paper is to establish the core principle that side information can serve as an operational approximation to nuisance structure to improve self-supervised learning; nevertheless, we have taken initial steps toward characterising these effects. In the controlled CIFAR experiments, Table 9 investigates the impact of imperfect side information by injecting controlled levels of label noise into the side dataset. These results show that the NF-MV mechanism remains effective even when the side information is only approximately nuisance-rich: performance degrades as noise increases. This demonstrates that the method does not require a perfectly curated nuisance distribution, and that some degree of overlap or contamination does not invalidate the NF-MV objective. However, we emphasize that this remains a controlled setting designed to isolate specific failure modes. In real-world applications, the notion of quality is inherently more complex. That said, we view a more thorough investigation of side information quality and scaling behaviour as an important direction for future work. The present results provide encouraging evidence that NF-MV is not brittle with respect to moderate imperfections in side information.

---

> ### Comment · Reviewer_NwSZ · 2025-11-25
>
> 1. The primary motivation of this paper is to address the limitations inherent in the conventional Multi-View assumption. While the authors identify meaningful challenges associated with this assumption and propose a Nuisance-Free Multi-View (NF-MV) assumption—reframing the objective of self-supervised learning (SSL) as learning representations that are sufficient for task-relevant information while remaining invariant to shared nuisance structure—this formulation still presents notable shortcomings.
>
> However, the NF-MV assumption relies heavily on domain expertise or supervised signals to accurately identify and disentangle side (nuisance) information from task-relevant information. In the absence of clear, practical guidelines for achieving separation, the proposed assumption lacks actionable methodological guidance. Consequently, it is difficult to empirically validate the central claim that the NF-MV assumption successfully achieves the stated goal of learning representations that are both sufficient for the target task and invariant to shared nuisance factors.
>
> 2. Many methods, such as BYOL, DINO, and VICReg, rely on the multi-view assumption. Your proposed Nuisance-Free Multi-View (NF-MV) approach can be applied to these methods as well. However, my original concern still hasn’t been addressed.
>
>
>
> My main concerns haven't been addressed, so I've lowered my rating to 4.

---

> ### Author Response · Authors · 2025-12-03
>
> We appreciate the Reviewer’s reply, yet we must confess to some confusion at the unexpectedly different assessment.
>
> ### 1. On the claim of "notable shortcomings".
> Given that no theoretical flaw in the NF-MV formulation was brought forward, while the perspective and framework it formulates is praised in the initial review, we struggle to understand this claim.
>
> The comment appears to refer to the "absence of clear, practical guidelines" on how side information is obtained. We are happy to clarify this aspect: a simple operational definition for side information is "background-only data". When domain expertise can identify features that are task-irrelevant but correlated with labels, incorporating this knowledge into the NF‑MV framework provides a safeguard against learning spurious dependencies that harm robustness and generalization. Our contributions are to prove this theoretically through our framework and to validate it experimentally. In our rebuttal, we described how side data is typically collected (and later discarded) as part of normal data acquisition processes.
>
> Crucially, the use of a side/auxiliary data to enhance learning of robust embeddings is not a novelty of our work, and is instead present in many other well-received works (e.g. Louiset et al., ICLR 2024; Weinberger et al., AISTATS 2022; Zou et al., NeurIPS 2013), viewing the ability to incorporate domain knowledge into self-supervised learning as a strength, not a limitation.
>
> ### 2. Empirical validation.
> To address the Reviewer's concern, we extended our experiments to additional SSL methods (BYOL, VICReg). As shown below, incorporating side information via our JSD penalty substantially improves robustness under spurious correlations for these additional methods as well, and show similar hyperparameter patterns:
>
> **Table:** Accuracy comparison between baselines and our approach with side information (-SI). The encoders are pre-trained on C-Cifar10: the LP/k-NN classifiers are either fitted with C-Cifar10 or U-Cifar10, and always validated on U-Cifar10 (spurious correlation removed).
>
> | Method     | γ   | LP: C→U | k-NN: C→U | LP: U→U     | k-NN: U→U    |
> |-----------|-----|---------|-----------|-------------|--------------|
> | VICReg    | --  | 49.64   | 44.82     | 79.20       | 64.00        |
> | VICReg-SI | 40  | 55.11   | 50.44     | 82.38       | 69.50        |
> | VICReg-SI | 80  | 59.94   | 55.03     | 84.27       | 73.08        |
> | VICReg-SI | 160 | 65.65   | 61.48     | **84.88**   | 76.06        |
> | VICReg-SI | 320 | **66.55** | **62.96** | 83.48     | **76.60**    |
> |           |     |         |           |             |              |  <!-- separator -->
> | BYOL      | --  | 53.46   | 43.05     | 83.28       | 74.23        |
> | BYOL-SI   | 2   | 58.05   | 49.84     | **84.35**   | 77.78    |
> | BYOL-SI   | 4   | 57.76   | 50.80     | 83.89       | **77.84**        |
> | BYOL-SI   | 8   | **58.90** | **53.04** | 83.12     | 77.54        |
>
> We truly hope these additional results have fully addressed any outstanding concerns regarding the applicability of NF-MV and the specific operational approach we have proposed. We will incorporate these results in the paper's appendix.

---

### Comment · Area_Chair_Jfi7 · 2025-11-23

Dear Reviewers,

The authors have submitted their rebuttal addressing your reviews. Please take the time to:

1. Read the rebuttal carefully
2. Ask clarifying questions if anything remains unclear
3. Update your scores and reviews based on the authors' responses

Please be mindful of timing: If you have follow-up questions for the authors, **post them early enough to give them adequate time to respond** before the discussion period closes on December 3rd.

Your timely engagement is crucial for a fair and thorough review process.

Thank you for your continued effort on this paper.

Best regards,
Area Chair

---

### Meta-Review · Area_Chair_EdMs · 2026-01-04

**Summary:**

Metareview: This work tackles an interesting problem, where multiview data in representation learning share both essential information and persistent, nuanced task-irrelevant information. Classical self-supervised and multiview learning approaches often overlook the latter. This work proposes an information-theoretic, side-information assisted approach to account for such settings.


Strengths:

The reviewers found that the setting is well motivated. They also found the paper easy to read and follow. The formalization of the nuance-free multiview assumption and the lightweight nature are also commended.


Weaknesses:

Some reviewers are unsatisfied by the small size experiments on MNIST and CIFAR, where the irrelevant information is easy to label. A reviewer pointed out that having expertise to label irrelevant information for the views appears to be unrealistic. More than one reviewer complained that there is a lack of guidance on selecting side information.

**Reviewer Concerns:**

The rebuttal argued that the MNIST and CIFAR settings were intentionally chosen to validate the theorem and the colonoscopy setting has much more complex nuances.

The rebuttal also acknowledged that using I(y:n)=0 to select the side information is an idealistic, not easy to implement criterion (mentioned in “limitations”), but argued the practice needs not to follow this information-theoretic criterion.

As AC, I found both responses are not entirely satisfactory. The guidance on how to label data with domain expertise is indeed unclear.

**Reviewer Scores:**

Reviewer NwSZ mentioned they would not change the rating. The major concern on heavy reliance of domain expertise was not addressed.

Reviewer 5eJ7 suggested to add comparison with medical SSL methods, change experiment resolution, and to add epsilon tolerance based theorem, but these were not addressed.

---

### Decision · Program_Chairs · 2026-01-26

Reject